



# Seasonality, drivers, and isotopic composition of soil $CO_2$ fluxes from tropical forests of the Congo Basin

Simon Baumgartner[1,5], Matti Barthel[1], Travis W. Drake[1], Marijn Bauters[2], Isaac Ahanamungu Makelele[2,3], John Kalume Mugula[3], Laura Summerauer[1], Nora Gallarotti[1], Landry Cizungu Ntaboba[4], Kristof Van Oost[5], Pascal Boeckx[2], Sebastian Doetterl[1], Roland A. Werner[1], and Johan Six[1]

[1]Department of Environmental Systems Science, Swiss Federal Institute of Technology, ETH Zurich, Switzerland
[2]Department of Green Chemistry and Technology, Ghent University, Belgium
[3]Département de Biologie, Université Officielle de Bukavu, DR Congo
[4]Department d'Agronomie, Université Catholique de Bukavu, DR Congo
[5]Earth and Life Institute, Université catholique de Louvain, Belgium

**Correspondence:** Simon Baumgartner, Earth and Life institute, Université Catholique de Louvain, Belgium (simon.baumgartner@uclouvain.be)

**Abstract.**

Soil respiration is an important carbon flux and key process determining the net ecosystem production of terrestrial ecosystems. To address the enormous lack of quantification and understanding of seasonality in soil respiration of tropical forests in the Congo Basin, soil $CO_2$ fluxes and potential controlling factors were measured for the first time annually in two dominant forest types (lowland and montane) of the Congo Basin during three years at varying temporal resolution. Soil $CO_2$ fluxes from the Congo Basin resulted in $3.69 \pm 1.22 \; and \; 3.82 \pm 1.15 \; \mu$mol $CO_2$ m$^{-2}$ s$^{-1}$ for lowland and montane forests, respectively. Respiration in montane forest soils showed a clear seasonality with decreasing flux rates during the dry season. Montane forest soil $CO_2$ fluxes were positively correlated with soil moisture while $CO_2$ fluxes in the lowland forest were not. Paired $\delta^{13}$C values of soil organic carbon (SOC) and soil $CO_2$ indicated that SOC in lowland forests is more decomposed than in montane forests, suggesting that respiration is controlled by C availability rather than environmental factors. In general, C in montane forests was more enriched in $^{13}$C throughout the whole cascade of carbon intake via photosynthesis, litterfall, SOC, and soil $CO_2$ compared to lowland forests, pointing to a more open system. Even though soil $CO_2$ fluxes are similarly high in lowland and montane forests of the Congo Basin, the drivers of them were different, i.e. soil moisture for montane forest and C availability for lowland forest.

## 1 Introduction

Soil basal respiration, the sum of carbon dioxide ($CO_2$) produced both autotrophically by roots and heterotrophically by microbial and fungal respiration, represents the biggest natural transfer of carbon (C) from the terrestrial biosphere to the atmosphere (Raich and Schlesinger, 1992). Globally, soil respiration is the second-largest terrestrial C flux after photosynthesis, emitting 98 Pg of C per year as $CO_2$ (Bond-Lamberty and Thomson, 2010). As such, the flux of $CO_2$ from soils represents





a significant component of net ecosystem production (NEP). Research into the abiotic and biotic controls of this flux are thus critical for understanding the overall C balance of ecosystems.

There are a number of different parameters that can influence soil $CO_2$ efflux, with soil temperature and soil moisture being the most important drivers (Ruehr, 2010). Soil temperature affects biological activity whereas soil moisture affects both the
respiration of organic matter and the diffusion of atmospheric oxygen and respired $CO_2$ through soil pores (Janssens et al., 1998; Doff Sotta et al., 2004; Sousa Neto et al., 2011; Courtois et al., 2018). In addition to temperature and moisture, soil pH (Courtois et al., 2018), through its effects on microbial communities, and root density can also affect soil $CO_2$ production (Janssens et al., 1998). Another particularly important driver is photosynthetic activity, as it describes the rate of carbohydrate supply from leaves to roots, where both root and rhizo-microbial respiration occur (Ekblad, 2001). While the magnitude of soil
respiration can vary significantly between and within different ecosystems, soil $CO_2$ fluxes from tropical forests are generally higher than from any other vegetation type, due to high soil temperature, high soil moisture (Raich et al., 2002), and weak C stabilization (Doetterl et al., 2018). This high soil $CO_2$ flux together with a high production leads to tropical forest having a fast C turnover. Nevertheless, as a whole the tropical terrestrial biosphere acts as a net C sink by production outbalancing the high soil respiration (Melillo et al., 1993; Pan et al., 2011; Palmer et al., 2019).

Despite the importance of tropical forests for the global C cycle, there is a lack of research into $CO_2$ fluxes from soils in these ecosystems. The Congo Basin in central Africa, hosts the second largest tropical forest on Earth, but has been particularly neglected in biogeochemical research. A recent paper on the greenhouse gas budget of Africa by Valentini et al. (2014) identified current key uncertainties and research gaps, especially on data availability of respiratory fluxes from tropical forests. This holds especially for the Congo basin, since there is only one study from 1962 reporting soil respiration rates from these forests
(Maldague and Hilger, 1962). Hence, while many studies model soil and ecosystem respiration in the Congo, there are almost no empirical data on soil $CO_2$ fluxes to validate the models. To model C fluxes in tropical Africa, these studies either upscale from a few FLUXNET stations or apply rates from other tropical forests (Jung et al., 2011). However, differences in species composition (Slik et al., 2015), forest structure (Lewis et al., 2013), nutrient atmospheric supply (Bauters et al., 2018), and climate patterns call for cross continental and spatially exhaustive monitoring across the tropics to fill up this important data gap
(Corlett, 2006). Eddy-covariance towers are the most common methods to measure $CO_2$ fluxes over different ecosystems and larger areas. However, continuous measurements of soil respiration close to the surface are needed to assess temporal trends of processes controlling soil $CO_2$ production and consumption (Ogle, 2018). This is particular important in light of recent data that show that the ratio of soil respiration to primary production has increased over time (Bond-Lamperty et al., 2018). In particular, heterotrophic respiration has increased as soil microbes became more active in response to increasing temperatures
(Bond-Lamperty et al., 2018). Bond-Lamberty and Thomson (2010) estimated that global soil respiration increased by 0.1 Pg C yr$^{-1}$ between 1989 and 2008, mostly due to an increase in air temperature. If this process proceeds such that ecosystem respiration exceeds primary production, terrestrial ecosystems could be transformed from sinks to sources of C. Thus, understanding baseline rates of soil respiration and the role of environmental drivers is crucial to assess future responses to climate change. This is especially important in the Congo Basin, as a recent study showed that the length of dry seasons have increased
by 6.4-10.4 days per decade since 1988 (Jiang et al., 2019). These changes in precipitation and temperature could trigger an





ecosystem response, including shifts in soil respiration. Furthermore, short-term events, such as extreme rain or prolonged dry periods, are predicted to occur more frequently with climate change and will most likely impact soil respiration rates (Hopkins and Del Prado, 2007; Borken and Matzner, 2009).

In light of these issues, the objectives of this study were to provide 1) the first empirical quantification of annual soil $CO_2$
fluxes from forests of the Congo Basin, 2) gauge variability between two dominant forest types within the basin, and 3) assess whether and to what extent soil temperature and moisture influence $CO_2$ fluxes. Soil respiration was measured weekly to assess the role of seasonality and environmental drivers of $CO_2$ fluxes. Additionally, stable C isotopic signatures ($\delta^{13}C$) of leaf-litter, soil organic carbon (SOC), soil-respired $CO_2$ and dissolved stream water $CO_2$, were measured to determine sources and sinks of soil respired $CO_2$.

## 2  Methods

### 2.1  Study sites

Old-growth forest sites in the Democratic Republic of Congo (DRC), contrasting in altitude, were selected to conduct long-term static manual chamber $CO_2$ flux measurements. The first site (KB) is situated in the Kahuzi-Biéga National Park (S 02.215°, E 28.759°) northwest of the city of Bukavu in the South-Kivu province and represents a montane tropical forest at
an altitude of 2120 m a.s.l with an annual mean temperature of 15 °C and an average annual rainfall of 1500 mm (Bauters et al., 2019). Rainfall peaks in both April and October, with a dry season from June to September in between (Alsdorf et al., 2016). The soils in KB are broadly classified as Umbric Ferralsols (Jones et al., 2013) with a sandy loam (upper 15 cm) to silt loam (15-30 cm) texture. The second site (YO) is situated in the Yoko Forest Reserve, south of the city of Kisangani in the Tshopo province (N 0.294°, E 25.302°). The YO site is a lowland tropical mixed forest with an annual average temperature of
24.2 °C and an average annual rainfall of 1800 mm (Bauters et al., 2019). The mixed forest plot is a classic African lowland rainforest with about 70 species per hectare and a canopy height up to 40 m (Doetterl et al., 2015; Kearsley et al., 2017). Like KB, there are two wet seasons, a short one from March to May and a longer one from August to November. The soils in YO are deeply weathered and nutrient poor Xanthic Ferralsols (Jones et al., 2013) with a loamy sand texture (0-30 cm). Because lowland forests are the main forest type within the Congo Basin, two additional lowland forest sites (Djolu and Yangambi)
were selected to conduct short term campaigns, assessing spatial robustness of the results (Figure 1). Yangambi is a UNESCO biosphere reserve and lies at the river bank of the Congo river about 100 km west of the city of Kisangani (Figure 1). Djolu is a territory just north of the equator roughly 300 km west of the city of Kisangani, in the north-east of Tshuapa province where measurements were conducted in protected forest areas (Figure 1). In Yangambi and Djolu measurements were conducted in old-growth mixed forest sites.

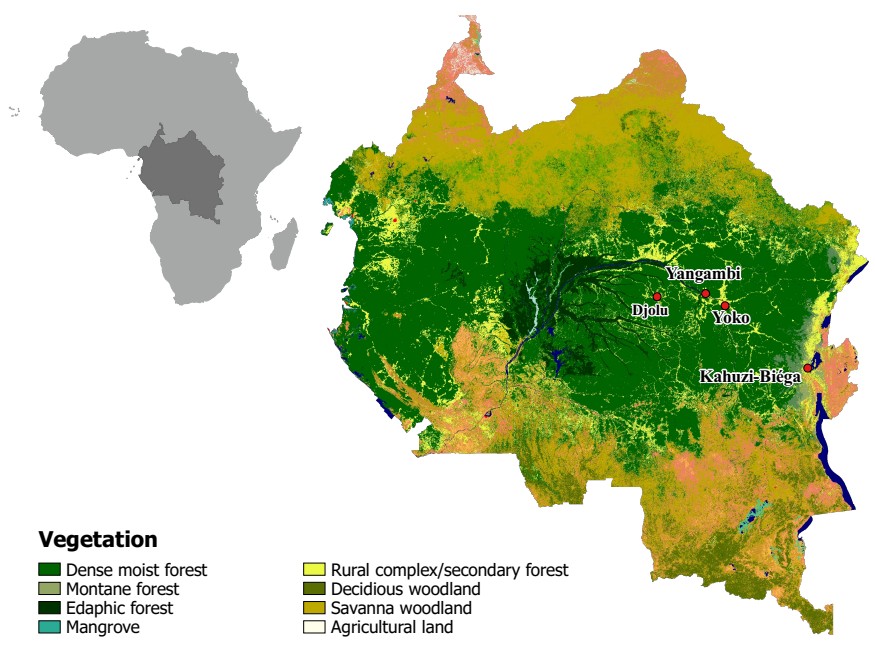

**Figure 1.** Map of part of the Congo Basin with the different vegetation types. Red dots indicate sampling locations. Lowland: Djolu, Yangambi, and Yoko. Montane: Kahuzi-Biéga. Map modified based on Verhegghen et al. (2012).

## 2.2 Soil $CO_2$ flux measurements

Flux chamber measurements were carried out at different time intervals during 2016-2019. Weekly to fortnightly sampling in YO was conducted from November 2016 to March 2019 and in KB from April 2017 to March 2019. In addition, several 2-week sampling campaigns with daily and sub-daily sampling were conducted to rule out diurnal soil respiration cycles (Figure

A3). These short-term sampling campaigns were conducted in KB (September 2016, April 2017), YO (October 2016, Mai 2017), Yangambi (September 2016) and in Djolu (May 2016). Sampling was done using the static manual chamber method, as described in Hutchinson and Mosier (1981). Briefly, at each site, a minimum of three PVC chambers with a diameter of 0.3 m, a height of 0.3 m, an airtight lid, and a vent tube to avoid pressure disturbances were installed. A thermocouple (Type T, Omega Engineering Deckenpfronn, Germany) was inserted through a gas tight cable gland to measure temperature in the chambers

at each sampling time point. Following established methods, the chambers were inserted into the forest floor at least 12 hours prior to taking the first sample to avoid altered results due to soil disturbance. The chambers remained in place throughout the measurement campaign. For each flux measurement, the lids of the chambers were closed and 20 mL headspace gas samples were withdrawn every 20 minutes (t0, t20, t40, and t60) with a 20 mL syringe. Gas samples were stored in pre-evacuated 12 mL vials with airtight septa (Exetainer; Labco Ltd, High Wycombe, UK). To avoid gas leakage issues as described by Knohl

et al. (2004), the septa were additionally sealed with a thin layer of silicon. To ensure that the headspace was well-mixed and that there was no static concentration gradient inside the chamber, the syringe was flushed with air from the chamber headspace





and reinjected into the chamber prior to sample withdrawal. Soil moisture probes (ECH$_2$O-5, Meter Environment, Pullman, U.S.) and air temperature data loggers (iButton, Maxim Integrated, San Jose, U.S.) were installed at each chamber cluster. Soil temperature was measured during each sampling event at 20 cm depth using a thermocouple (Type T, Omega Engineering, Deckenpfronn, Germany). To standardize soil moisture data between sites and soil types, the water filled pore space (WFPS)

was calculated for each volumetric water content measurement using bulk soil density data provided from Bauters et al. (2019) and particle density of soil minerals of 2.65 g cm$^{-2}$.

Gas samples were analyzed for concentrations of CO$_2$ at ETH Zurich (Zurich, Switzerland) using gas chromatography (Bruker, 456-GC, Scion Instruments, Livingstone, U.K.). Soil gas fluxes were calculated using the linear increase of the gas concentration in the head space of the chambers over time, corrected for pressure and temperature according to the ideal

gas law, divided by chamber area (Imer et al., 2013). Using the micrometeorological convention, a flux from the soil to the atmosphere is denoted as positive flux.

## 2.3  $\delta^{13}$C-CO$_2$ of streams and soil respiration

After concentration measurements, the remaining gas was analyzed for $\delta^{13}$C of CO$_2$ for one week of each month and site to derive a representative $\delta^{13}$C signature of the monthly soil-derived CO$_2$ via the Keeling plot approach (Keeling, 1958). All

Keeling plots yielded an r$^2$ > 0.99 (Fig A1). Post-run off-line calculation and drift correction for assigning the final $\delta^{13}$C values on the V-PDB scale were done following the "IT principle" as described by Werner and Brand (2001). The $\delta^{13}$C-values of the laboratory air standards were determined at Max-Planck-Institute for Biogeochemistry (Jena, Germany) according to Werner et al. (2001). Briefly, linking measured $\delta^{13}$C values of CO$_2$ gas isolated from standard air samples relative to the carbonate V-PDB scale was done via the Jena Reference Air Standard (JRAS), perfectly suited to serve as a primary scale anchor for

CO$_2$-in-air measurements. The measurement of the aliquots of the laboratory standards is routineously better than 0.15‰. In addition to soil CO$_2$, dissolved CO$_2$ samples of six pristine headwater streams near the chamber sites were taken in April (wet season) and September 2018 (dry season) using the headspace equilibration technique. At each stream site, 20 mL of unfiltered water sampled from the thalweg was injected into 110mL, N$_2$-flushed (Alphagaz 2, Carbagas, Gümlingen, Switzerland) serum crimp vials containing 50 $\mu$L of 50 % ZnCl. From the headspace of the crimp vials, three analytical replicates were subsampled

into evacuated 12mL exetainers (Labco Limited, High Wycombe, UK) following Bastviken et al. (2008). According to Szaran (1998) only 1.03 permille fractionation occurs between dissolved and gas phase, thus $\delta^{13}$C of headspace CO$_2$ can be used as a representative measure for dissolved $\delta^{13}$C of CO$_2$. All CO$_2$ samples were analyzed for $\delta^{13}$C of CO$_2$ with a modified Gasbench II periphery (Finnigan MAT, Bremen, D) coupled to an isotope ratio mass spectrometer (IRMS; Delta$^{plus}$XP; Finnigan MAT; modification as described by Zeeman et al. (2008)) (see Supplemental Information).

## 2.4  $\delta^{13}$C of litter and soil


Litterfall collected fortnightly between 2015 and 2016 from traps installed at the same sites was used to determine $\delta^{13}$C of leaves. At each site, the leaves were combined into monthly samples which were subsequently dried, homogenized, and ground (Bauters et al., 2019). Soil samples were taken at the montane and the lowland forest plots at 0-30 cm depth and subsequently





air dried, sieved and milled. Litter and soil samples were analyzed using elemental analyzer (Automated Nitrogen Carbon Analyser; ANCA-SL, SerCon, UK), interfaced with an Isotope Ratios Mass Spectrometer (IRMS; 20-20, SerCon, UK).

### 2.5 Statistical Analyses

In total 1108 single flux measurements have been conducted (398 in the montane forest and 710 in the lowland forests, respec-
tively). As the campaigns from the different sites were spread over several years, all data were compiled and averaged into weekly bins prior to plotting time series of the data assuming little year-to-year variability. In that way, yearly site averages were not weighed by periods of intensive sampling as each single week had an assigned median value regardless of measurement frequency. Effects of forest type, soil temperature and WFPS on the soil $CO_2$ flux were quantified using a linear mixed effects model, including all fluxes that were measured, and controlling for the soil chamber via a random intercept. Because
a full model was not converging for soil $CO_2$ flux, including all interaction terms between the three predictors, interaction between WFPS and soil temperature were omitted. Likewise, a model was run to explain effects of forest type, ecosystem compartment (litter, soil $CO_2$ flux and stream $CO_2$) and season (wet/dry season) on $\delta^{13}C$ values, including sample spot (litter-fall trap, soil flux chamber and sampled stream) as a random effect. Models were fitted using maximum likelihood methods via the lmerTest package (Kuznetsova et al., 2017). P-values of the fixed effects - elevation, transect and their interaction - were
determined based on the denominator degrees of freedom calculated with the Satterhwaite approximation. Marginal (m) and conditional (c) $R^2_{adj}$ are proxies for the variation explained by the fixed effects, and both the random and fixed effects, respectively, were calculated following Nakagawa and Schielzeth (2013), via the MuMIn package (Barton, 2019). For all statistical analyses, the R-software was used (R Development Core Team, 2019). All model fits were validated by checking normality and homoscedasticity of the residuals. QGIS version 2.18 was used to compile the map of the Congo Basin.

## 3 Results

### 3.1 Temperatures and Soil Moisture

Weekly mean soil and air temperature were both stable throughout the year in both forest types (Figure 2a). Average soil temperatures were 24.0 and 15.3 °C in the lowland and montane forest sites, respectively. Air temperatures were slightly lower in both lowland and montane sites, averaging 23.5 and 14.7 °C, respectively. The WFPS at 30 cm depth in the lowland forest
was quite constant. However, a decrease in WFPS was observed during dry season in the montane forest (Figure 2b). Annual average WFPS in the montane forest was higher (51.4 %) than in the lowland (29.6 %).

### 3.2 Soil $CO_2$ fluxes

Average annual soil $CO_2$ fluxes were (average $^{maximum}_{minimum} \pm SD$) $3.69\,^{9.64}_{1.30} \pm 1.22$ $\mu$mol m$^{-2}$ s$^{-1}$ and $3.82\,^{6.33}_{0.42} \pm 1.15$ $\mu$mol m$^{-2}$ s$^{-1}$ for the lowland and montane forests, respectively. Soil $CO_2$ fluxes in the montane forests were lowest during the
dry season from June to September ($2.95\,^{5.44}_{1.25} \pm 0.95$ $\mu$mol m$^{-2}$ s$^{-1}$) and highest during the wet season during October to

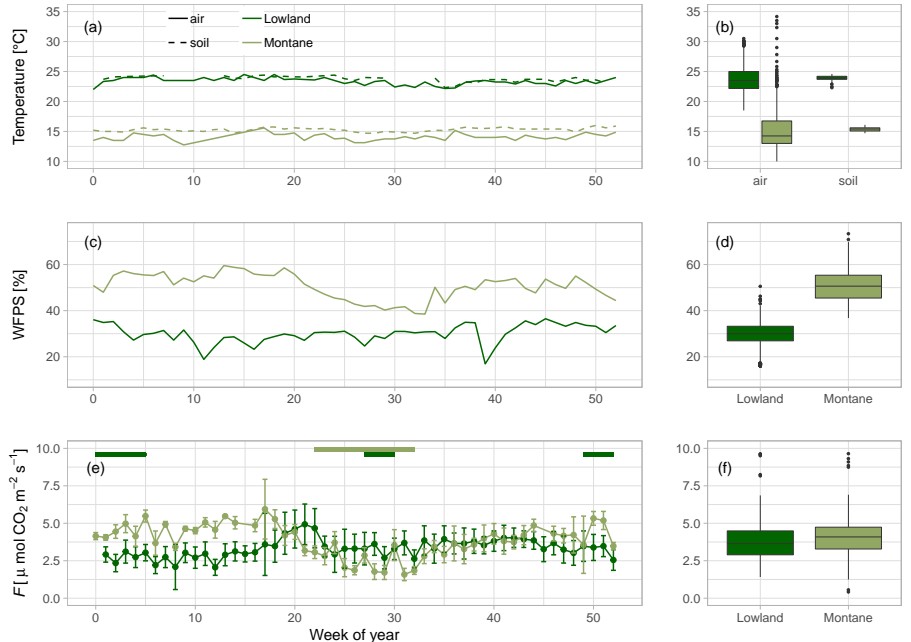

**Figure 2.** (a) Weekly averaged air (solid line) and soil temperature at 20 cm depth (dashed line) in the lowland (dark green) and montane (light green) forest sites. (b) Yearly median air- and soil temperature. (c) Weekly average water-filled pore space (WFPS) [%] in the lowland and montane forest soils at 30 cm depth. (d) Median WFPS in the lowland and montane forest. (e) Weekly median soil $CO_2$ fluxes ($F$) with error bars indicating the standard deviation. Green horizontal bars on top of panel C indicate the dry seasons in the lowland and in the montane forests, respectively. (f) Median soil $CO_2$ fluxes i the lowland and montane forests.

May (4.49 $^{6.33}_{0.42}$ ± 0.79 $\mu$mol m$^{-2}$ s$^{-1}$) (Figure 2c). Lowland fluxes were more stable throughout the year, with only a small increase at the end of the wet season in June (4.67 $^{6.12}_{2.04}$ ± 1.31 $\mu$mol m$^{-2}$ s$^{-1}$) (Figure 2c). The linear mixed-effect model for soil $CO_2$ flux explained 68 % of the overall variability, with 40 % allocated to fixed effects (forest type, soil temperature and WFPS) (Table 1).The linear mixed effect model showed a negative effect of soil temperature on soil $CO_2$ flux in the lowlands

5 (P-value < 0.01) but a positive effect in the montane forest (P-value <0.001). In montane forest, a positive effect of WFPS on soil $CO_2$ flux (P-value <0.001) was observed (Table 1).

### 3.3 $\delta^{13}$C values of leaf, litter, soil respired $CO_2$, and dissolved $CO_2$ in headwater streams

For each category (litter, soil, soil $CO_2$, stream $CO_2$), the $\delta^{13}$C values in the lowland sites were always more negative than in the montane forest (Figure 3). The most negative values are found in leaf litter (-29.91 ± 0.94 and -28.56 ± 0.85‰ for

10 lowland and montane, respectively). The highest values were found in stream dissolved $CO_2$, with -22.74 ± 2.34 and -16.68 ± 0.95‰ in lowland and montane streams, respectively. In both forest types, the $\delta^{13}$C values increased from litter via SOC and soil respired $CO_2$ to dissolved $CO_2$ in streams. Only soil $\delta^{13}$C-$CO_2$ in the montane forest showed a small decrease relative





**Table 1.** Fixed effects estimates for both the $CO_2$ flux and the $\delta^{13}C$ response variables, including 1) forest type (lowland - montane), water-filled pore space (WFPS in %), soil temperature (in °C) and their interactions as predictors for the soil $CO_2$ efflux (in $\mu$mol m$^{-2}$ s$^{-1}$), and 2) forest type (lowland - montane), ecosystem compartment (litter - soil $CO_2$ - stream $CO_2$) and season (wet - dry season) and their interaction as predictors for the $\delta^{13}C$ values (in ‰). For each effect, estimated standard error and estimated P-values is given, along with the estimated marginal (m) and conditional (c) $R^2_{adj}$ (Nakagawa and Schielzeth, 2013)

| Response | Effect | Estimate | SE | P-value | $R^2_{adj,m}$ | $R^2_{adj,c}$ |
|---|---|---|---|---|---|---|
| $CO_2$ flux | Intercept: Lowland forest | 10.074 | 2.37 | <0.001 | 0.4 | 0.68 |
| | Montane forest | -24.104 | 0.01 | 0.88 | | |
| | WFPS | -0.001 | 3.26 | < 0.001 | | |
| | Soil temperature | -0.280 | 0.10 | <0.01 | | |
| | WFPS: Montane forest | 0.137 | 0.01 | <0.001 | | |
| | Soil temperature: Montane forest | 1.005 | 0.19 | <0.001 | | |
| $\delta^{13}C$ | Intercept: Lowland forest - Litter - Dry season | -29.951 | 0.61 | <0.001 | 0.9 | 0.97 |
| | Montane forest | 1.018 | 1.23 | 0.42 | | |
| | Wet season | 0.042 | 0.23 | 0.85 | | |
| | Soil $CO_2$ | 1.469 | 0.78 | 0.08 | | |
| | Stream $CO_2$ | 7.737 | 0.88 | < 0.001 | | |
| | Montane forest - Wet season | 0.510 | 0.50 | 0.31 | | |
| | Montane forest - Soil $CO_2$ | 1.773 | 1.45 | 0.24 | | |
| | Montane forest - Stream $CO_2$ | 5.070 | 1.51 | <0.01 | | |
| | Wet season - Soil $CO_2$ | -0.104 | 0.32 | 0.75 | | |
| | Wet season - Stream $CO_2$ | -1.101 | 0.42 | 0.01 | | |
| | Montane forest - Wet season - Soil $CO_2$ | -1.582 | 0.60 | 0.01 | | |
| | Montane forest - Wet season - Stream $CO_2$ | -0.567 | 0.70 | 0.42 | | |

to soil C (Figure 3). Monthly leaf litter $\delta^{13}C$ did not show temporal variability (Figure A2d). The average $\delta^{13}C$ value of soil respired $CO_2$ was -28.35 $\pm$ 0.58 and -26.39 $\pm$ 1.03‰ in the lowlands and montane forests, respectively. The linear mixed model showed a statistical difference in the $\delta^{13}C$ values of soil $CO_2$ in the montane forest between the wet and dry season, however, no difference was found in the lowland forest. Whereas there was no significant difference in $^{13}C$ values of dissolved $CO_2$ between wet and dry season in the montane streams, a significant depletion in $^{13}C$ in the wet season in lowland streams was found (Table 1).



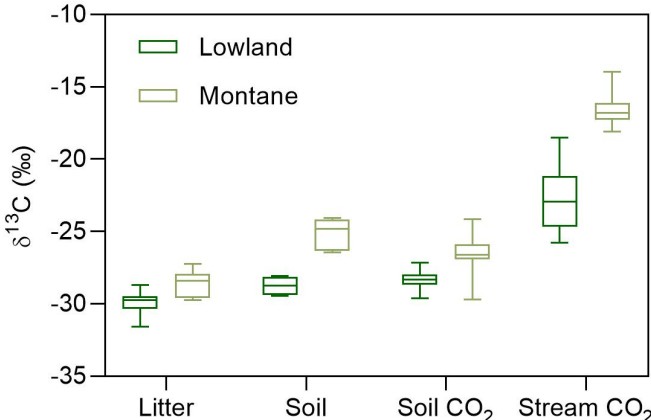

**Figure 3.** $\delta^{13}C$ values of litter, soil organic carbon, soil respired $CO_2$ and dissolved $CO_2$ in headwater streams adjacent to the monitoring sites in the lowland and montane forests.

## 4 Discussion

### 4.1 Soil $CO_2$ Fluxes

Long term studies of soil $CO_2$ fluxes in tropical forests are scarce, especially in the Congo Basin. The high temporal resolution data presented here, with 1108 individual soil $CO_2$ flux measurements over a period of more than 2 years, represents the

most exhaustive dataset for the Congo Basin. The results from the intense sampling campaigns showed that there is not a big variability in soil $CO_2$ fluxes between different lowland forest sites (Figure A3 b) and that fluxes are stable within a site and throughout a day (Figure A3 d). The average annual values reported in this study of $3.82 \pm 1.15\ \mu mol\ m^{-2}\ s^{-1}$ and $3.69 \pm 1.22\ \mu mol\ m^{-2}\ s^{-1}$ for the montane and lowland forests in the Congo Basin (Figure 2f), respectively, are within the range of reported values from other tropical forests. It is, reported average soil $CO_2$ fluxes from South and Central American tropical

forests were for French Guiana (2.30 to $5.30\ \mu mol\ m^{-2}\ s^{-1}$, (Buchmann et al., 1997; Janssens et al., 1998; Bréchet et al., 2011; Epron et al., 2013; Courtois et al., 2018)), Brazil (2.64 to $4.30\ \mu mol\ m^{-2}\ s^{-1}$, (Davidson et al., 2004; Doff Sotta et al., 2004; Sousa Neto et al., 2011; Sotta et al., 2007; Garcia-Montiel et al., 2004)), and Panama ($5.20\ \mu mol\ m^{-2}\ s^{-1}$, (Pendall et al., 2010)). To our knowledge, the only reported soil $CO_2$ fluxes from a tropical forest in Africa in recent years are from Kenya (Arias-Navarro et al., 2017; Werner et al., 2007) and they were compared to our flux rates rather low (i.e., between 1.04 and

$1.66\ \mu mol\ m^{-2}\ s^{-1}$). Higher fluxes were reported in tropical forests in Hawaii ($6.96\ \mu mol\ m^{-2}\ s^{-1}$, (Townsend et al., 1995)) and Thailand ($9.76\ \mu mol\ m^{-2}\ s^{-2}$, (Hashimoto et al., 2004)). The data presented here are the first from tropical forests within the Congo Basin since Maldague and Hilger (1962) reported soil respiration values from lowland forests in the DR Congo of 3 to $4\ \mu mol\ m^{-2}\ s^{-1}$. These values from the year 1962 lie exactly within the range of the values measured in this study, although it is important to note that the 1962 fluxes were derived from only four observations.





## 4.2 Seasonality of $CO_2$ Fluxes

Fluxes in the montane forest showed marked seasonality with a 34 % decrease during the dry season whereas fluxes in the lowland forests did not show any seasonality (Figure 2c). Courtois et al. (2018) have shown a similar trend of decreased fluxes during the dry season (15.7 % decrease) in tropical forests in French Guiana, however they were not as pronounced as in the

montane forests we sampled. One possible reason for the seasonal difference between montane and lowland forests is that the lowland dry season is not as distinct as in the montane regions. Rainfall events during the "dry season" in the lowlands are not uncommon (Figure A2). A model study by Raich et al. (2002) concluded that in seasonally dry biomes, soil $CO_2$ emissions positively correlate with precipitation. Precipitation was also identified as the main driver of maximum C assimilation rates in 11 Sub-Saharan ecosystems, which in turn was an ultimate driver of soil $CO_2$ fluxes (Merbold et al., 2009). Thus, given the

results of the present study and the projected increase in dry season length in the Congo Basin, as recently reported by Jiang et al. (2019), one would expect a future decrease of C fluxes in the montane forests while little to no effect might be expected in the lowland forests.

## 4.3 Temperature and Soil Moisture Controls

Despite the markedly different temperature regimes between the lowland and montane forests, yearly averaged soil $CO_2$ fluxes

were almost identical (Figure 2 f). Such inter-site temperature independence of soil $CO_2$ flux is unique compared to other tropical (e.g. Brazil (Doff Sotta et al., 2004; Sousa Neto et al., 2011) or temperate forests (e.g. Switzerland (Ruehr, 2010)), where strong correlations between soil $CO_2$ fluxes and soil temperature were found. However, in addition to temperature, soil geochemistry can play a crucial role in soil respiration, particularly via soil C stabilization processes and their rates (Doetterl et al., 2018). Short-term changes in C fluxes are mostly related to respiration of non-protected soil C (plant residues, root

exudates, rhizodeposition) while the majority of stored C in soils is stabilized within the mineral matrix (Doetterl et al., 2018). Thus, a potential increase in soil respiration due to change in soil temperatures in the lowland might be counteracted by higher C protection due to soil geochemistry. Within sites, a statistical significant correlation between soil temperature and $CO_2$ flux was found, however, montane and lowland forest displayed opposite relationships with soil temperature (Table 1). The negative relationship with soil temperature in the lowland forest indicates that soil temperatures are already too high for an

optimal microbial activity. Nevertheless, it is important to note that the soil temperatures within different forest types of the Congo Basin are relatively stable throughout the year (Figure 2 a), with standard deviations of 0.34°C and 0.42°C for montane and lowland soil temperature, respectively. Thus, given the lack of variability in soil temperatures, the accuracy of detected relationships could be questioned and extrapolations of the here found $CO_2$ flux responses beyond the soil temperature ranges observed in this study should be handled with care. For better understanding of temperature dependencies of soil respiration in

these forest soils, warming experiments (incubation or field) are needed.

Despite the higher total annual rainfall in the lowlands (Figure A2), the montane forest soils exhibited higher WFPS (Figure 2 b). The lower relative WFPS in the lowlands was likely due to the sandier soil texture leading to faster drainage. Moreover, the montane forest site showed a clear positive relationship between soil $CO_2$ and WFPS, whereas in the lowland site soil





$CO_2$ flux was uncorrelated with WFPS (Table 1). Soil moisture can influence soil respiration physically and biologically. Physically, soil moisture can limit the diffusion of gases through soils, including both oxygen required for aerobic respiration and respiratory $CO_2$. Biologically, soil moisture can affect the activity of heterotropic respiration, where low soil moisture conditions stress soil microbial communities and autotrophic respiration (Xu and Qi, 2001; Rey et al., 2002). The latter is

linked to canopy processes, where water limitation can lead to stomatal closure, limiting plant photosynthesis and thus also belowground respiratory processes (see also 4.4). One possible explanation for why the lowland soils $CO_2$ flux did not vary with WFPS is that the soil respiration is limited by soil C availability, indicated by the similar isotope composition of litter, SOC and soil emitted $CO_2$ in the lowland forests (see also 4.4). Therefore, if soil respiration in lowland forests is indeed likely substrate limited, then environmental factors such as soil moisture or temperature may have less control on soil respiration (Davidson

and Janssens, 2006). While lowland forests are most likely C-limited, montane forests are more sensitive to environmental conditions and represent a potentially large C source with climate warming.

### 4.4   Isotopic source indicators

It is well known that soil respiration and canopy processes are linked in forests (Ekblad, 2001). Carbohydrates produced by photosynthesis are subsequently transported to the roots and rhizosphere, where they are respired by root or microbial respira-

tion (Ruehr et al., 2009). Thus, the isotopic signature of soil-derived $CO_2$ is mostly governed by isotope fractionation processes that occur at the leaf scale, since a significant portion of soil respired $CO_2$ is supplied by recent photoassimilates (Högberg et al., 2001; Brüggemann et al., 2011; Barthel et al., 2011). Generally, all environmental parameters affecting photosynthesis and thus $CO_2$ discrimination are likely to influence the $\delta^{13}C$ signal of soil respiration (e.g. precipitation, vapor pressure deficit) (Bowling et al.). In this study, the link between photosynthesis and soil $CO_2$ is evident through the distinctively different $\delta^{13}C$

signatures between lowland and montane forests along the whole cascade of carbon intake via photosynthesis, litterfall, SOC, soil $CO_2$ and stream dissolved $CO_2$. This difference holds throughout most of the year for leaf litter and soil respired $CO_2$ between the lowland and the montane site (Figure A2). The strongest $^{13}C$ enrichment of soil $CO_2$ was observed at the end of the dry season (September) in the montane site, likely caused by lower photosynthetic $^{13}C$-$CO_2$ discrimination conveyed to soil respiration (Figure A2 c). Indeed, the enrichment of $^{13}C$ of autotrophic soil respiration resulting from stomatal closure

during periods of drought has been widely documented (Ekblad, 2001; McDowell et al., 2004; Blessing et al., 2016; Salmon et al., 2019). Such distinct enrichment was presumably not detected in the lowland sites due to the absence of a prolonged dry season (Figure A2 a). A study by Ometto et al. (2002) found similar seasonal dependencies of two tropical forest of the Amazon Basin, with one site (Santarem forest) showing a distinct seasonality of $\delta^{13}C$ signal of ecosystem respiration in response to large variation in rainfall whereas the other (Manaus forest) had only little variation in rainfall and thus also little variation

in $\delta^{13}C$ signal of ecosystem respiration.

The $\delta^{13}C$ value of various ecosystem components (leaf litter, SOC, soil respired $CO_2$, and riverine $CO_2$) were universally enriched in the montane compared to lowland forests (Figure 3). Similar isotopic enrichment with altitude has been shown even within small-scale gradients in Amazonian forests (de Araújo et al., 2008). In the Amazonian study, the relatively enriched values of leaf and ecosystem respiration in the high elevation sites was explained by increased leaf-level photosynthetic capacity





(higher leaf nitrogen content and leaf mass per unit area (LMA)), which is decreasing intercellular $CO_2$ concentrations and reducing leaf discrimination, resulting in increasing $^{13}C$ concentrations in the leaves (de Araújo et al., 2008). However, Bauters et al. (2017) reported decreasing leaf nitrogen content and LMA with higher elevations in tropical forests of the Congo Basin. It is more likely, that the higher $\delta^{13}C$ values in the montane forest are linked to canopy processes and lower temperatures. A shift

in microbial communities due to temperature changes has been found to impact fractionation of the C isotope in heterotrophic soil respiration (enrichment of $^{13}C$ at lower temperatures) (Andrews et al., 2000). Overall, different $\delta^{13}C$ values in the studied ecosystem components between the two forest types might be due to a combination of different effects including temperature, canopy processes, and open vs. closed system isotope dynamics.

As C is respired and transferred down the cascade from photosynthesis to stream dissolved $CO_2$, it becomes more enriched

with heavier isotopes when transiting from one pool to the next due to isotope fractionation (as $^{12}C$ tends to be preferentially consumed). This is generally a feature of 'open systems' in which reactions occur with a continuous supply of substrate, while the residual substrate and products are lost from the system. In contrast, a 'closed system' is characterized by the absence of new inputs and results in less fractionation between substrate and product (Fry, 2006). The different enrichment gradients observed between lowland and montane tropical forests indicate typical closed vs. open system dynamics, respectively. In

particular, the similar isotopic signatures of litter, SOC, and emitted soil $CO_2$ at the lowland site indicated a more complete decomposition of the C input into the different compartments and thus closed system isotope dynamics (Figure 3). Moreover, in both systems, the highest enrichment occurs in the terminal location as stream dissolved $CO_2$. A similar enrichment of stream $CO_2$ relative to soil respired $CO_2$ has also been found in the Amazon by Quay et al. (1989). However, since stream $CO_2$ is governed by a multitude of factors (Enrichment factors: aquatic photosynthesis, equilibration with atmosphere, outgassing,

and weathering of carbonate/silicate minerals (depends on $CO_2$ source for $SiO_2$); Depletion factors: respiration of organic C, possibly photodegradation) it remains difficult to isolate a single factor causing the different isotope effects between soil $CO_2$ and stream dissolved $CO_2$ for lowland and montane forest.

## 5  Conclusions

Although the lowland and montane forests of the Congo Basin differed in terms of vegetation composition, climate, and edaphic

conditions, there was no significant difference in annually averaged soil $CO_2$ flux observed in this study. The montane forest site exhibited strong seasonality, primarily driven by water filled pore space. It was not possible to asses temperature dependency within a site, as the temperature range was too small. Furthermore, no temperature dependency between sites was found. $\delta^{13}C$ signatures exhibited a relative enrichment in montane site compared to the lowland across various ecosystem components (leaf litter, SOC, soil respired $CO_2$, and riverine $CO_2$). The montane forest also uniquely displayed seasonal variations of $\delta^{13}C$

signal of soil respired $CO_2$, likely driven by changing discrimination at the canopy scale. In contrast, the nearly identical C isotopic signatures of soil derived $CO_2$, litter, and SOC in the lowland forest indicate that respiration is likely substrate limited. Substrate limitation in the lowlands would also limit the influence of environmental factors such as WFPS on the $CO_2$ flux rate, which corresponds well to the observed lack of correlation between soil moisture or temperature with soil $CO_2$ fluxes. Overall,



these results fill a critical knowledge gap for soil respiration rates of major tropical forests, provide baseline flux magnitudes to better parameterize earth system models, and highlight how soil respiration in montane tropical forest soils of the Congo Basin are relatively sensitive to environmental changes and may become significant source of C to the atmosphere under a warming climate regime.

*Data availability.* All data used in this study were published at Zenodo and are available under http://doi.org/10.5281/zenodo.3757768.

## Appendix A: Method supplement

### A1 $\delta^{13}$C measurement of air samples with the Gasbench

Carbon isotopic composition of $CO_2$ in gaseous samples were measured with a modified Gasbench II periphery (Finnigan MAT, Bremen, D) coupled to an isotope ratio mass spectrometer (Delta$^{plus}$XP; Finnigan MAT; modification as described

by Zeeman et al. (2008)). In short, the modification of the Gasbench comprises the replacement of the GC-type split by a ConFlo III-like split and the addition of a home-built cold trap (1/10" SS capillary filled with Ni-wire, Goodfellow GmbH, Bad Nauheim, D) instead of the standard sample loop of the 8-port valve inside the Gasbench II. The gas mixture in the exetainer is transferred to the cold trap after piercing the septum with a vendor-supplied double-holed needle connected to two capillaries (fused silica and 1/32" steel capillaries). The feed capillary delivers pure He allowing a pressure build-up in the exetainer which

flushes the sample gas at a rate of about 0.5 mL/min over Nafion dryers to the cold trap where condensible gases (mainly $CO_2$ and $N_2O$) are frozen out with liquid nitrogen. The cold trap is connected to a pressurized Dewar vessel and equipped with a computer-controlled automatic refill unit (Zeeman et al., 2008) allowing automatically refilling of the cold-trap. After diverting the non-consensible gases to a vent, the cold trap is thawed and the content of the cold trap is automatically injected on a GC column (Poraplot Q 25 m x 320 mm i.d. (Varian, Walnut Creek, USA) held at 24°C) to allow separation of the isobar gases

$CO_2$ and $N_2O$. Post-run off-line calculation and drift correction for assigning the final 13C values on the V-PDB scale were done following the "IT principle" as described by Werner and Brand (2001). The $\delta^{13}$C- (and $\delta^{18}$O-) values of the laboratory air standards were determined at the Max-Plack-Institute for Biogeochemstry (Jena, D) according to Werner et al. (2001). The linking of the measured $\delta^{13}$C (and $\delta^{18}$O) values of $CO_2$ gas isolated from air samples relative to the carbonate V-PDB scale was done via the Jena Reference Air Standard (JRAS), perfectly suited to serve as a primary scale anchor for $CO_2$–in-air

measurements The measurement of the aliquots of the laboratory standards is routineously better than 0.15 ‰.

## Appendix B: Results supplement





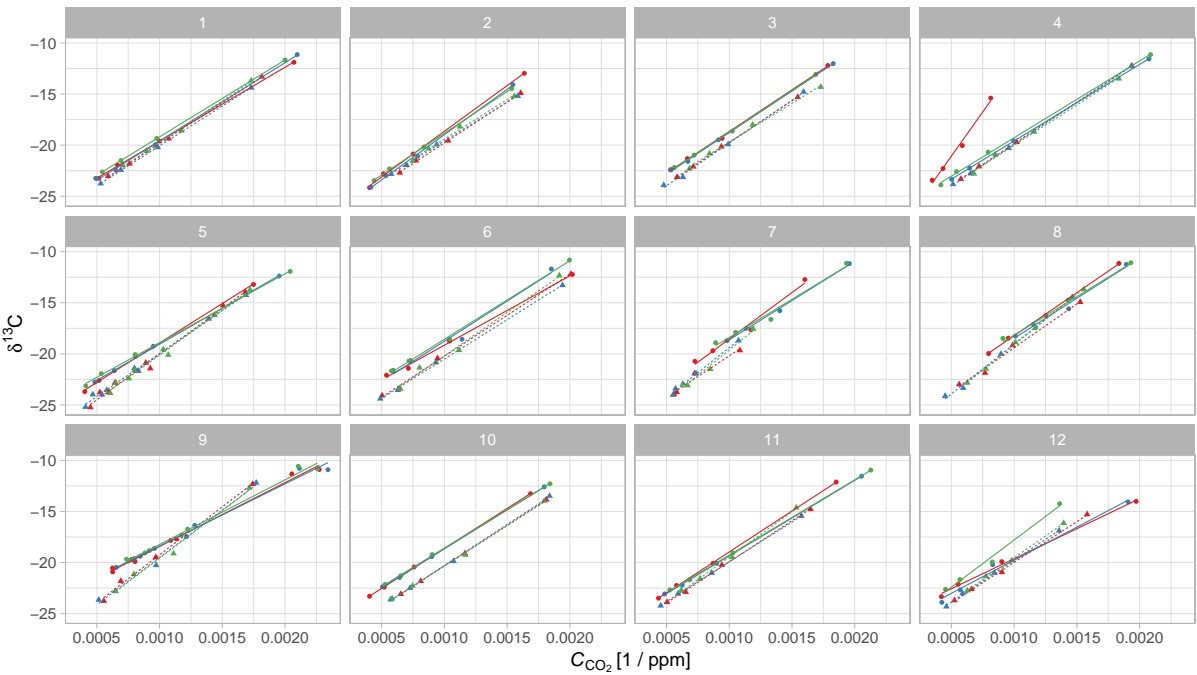

**Figure A1.** Shown are all individual Keeling plots for each chamber replicate (colour, n = 3) per site and per month of Kahuzi-Biéga (circles, solid lines) and Yoko (triangles, dashed lines) forest sites.

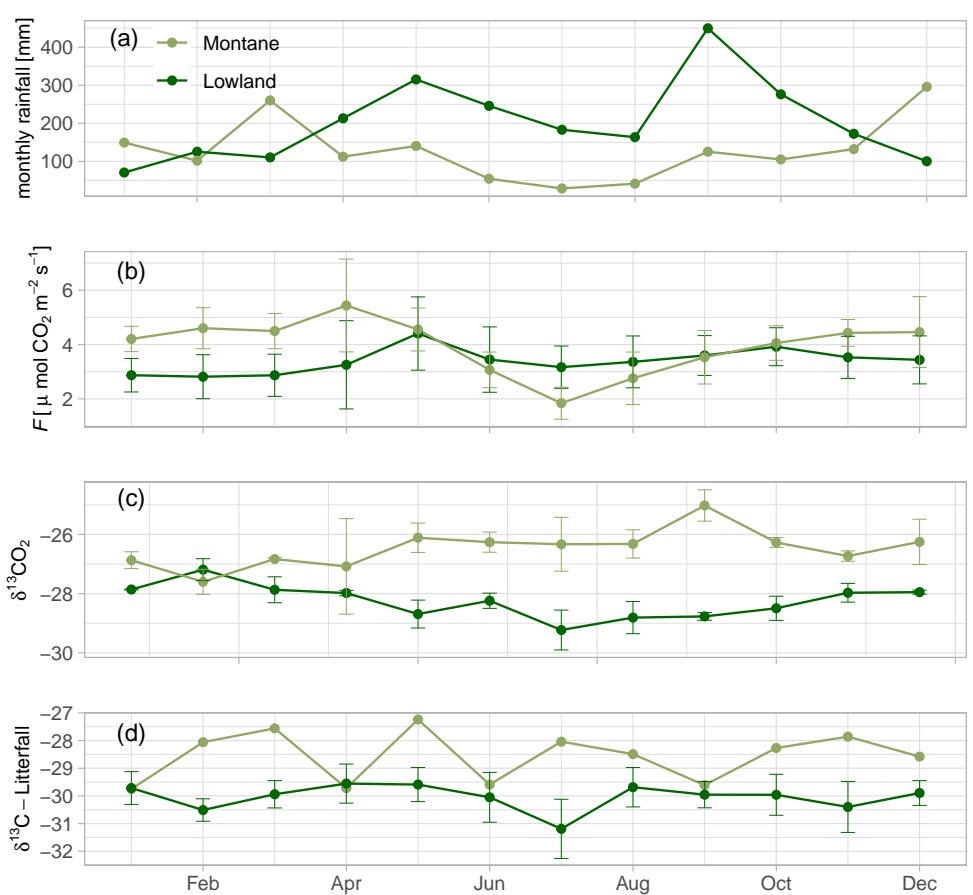

**Figure A2.** a) Monthly rainfall in mm at a lowland site in Yangambi and in Bukavu near the montane site. b) Monthly median $CO_2$ fluxes in the lowland and montane forests. c) Monthly median $\delta^{13}C$ values of the soil respired $CO_2$. d) Monthly $\delta^{13}C$ of litter in montane and lowland forests. Error bars indicating standard deviation.



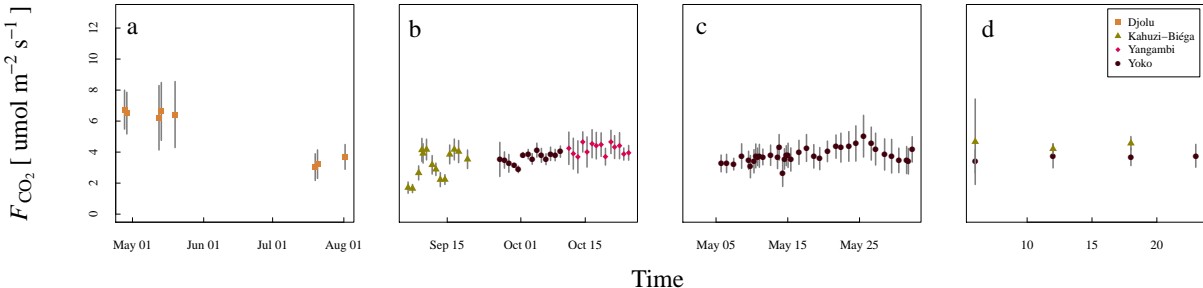

**Figure A3.** Median CO$_2$ fluxes with errorbars indicating standard deviation. a) Sampling campaign in a lowland forest in Djolu between May and August 2016. b) Sampling campaign in Kahuzi-Biéga (montane forest) Yoko and Yangambi (lowland forests) in September and October 2016. c) Sampling campaign in Yoko in May 2017. d) Sub-daily sampling in Kahuzi-Biéga and Yoko. x-Axis shows the hour of the day.

*Author contributions.* SB, MBarthel and JS were responsible for study design. Fieldwork was conducted by SB, MBarthel, IAM, JKM, LS, and NG. Lab work was conducted by MBarthel and RAW. Data interpretation was performed by SB, MBarthel, TWD, MBauters supported by KVO, PB and JS. The manuscript was written by SB with contributions from all co-authors.

*Competing interests.* The authors declare no conflict of interest.

5  *Acknowledgements.* We want to thank Héritier Ololo Fundji and Montfort Bagalwa Rukeza for administrative and logistic support during the field campaigns in the DRC; the park authorities of Kahuzi-Biéga National Park, the INERA in Yangambi for hosting our study, and the park guards for security. The study was financed by core funding of ETH Zurich and the Swiss National Science Foundation (SNSF). SB is currently financed by core funding of ETH Zurich and the Fonds National de la Recherche Scientifique (FNRS).



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
