# Peer review of "Seasonality, drivers, and isotopic composition of soil CO2 fluxes from tropical forests of the Congo Basin"

_Biogeosciences, 2020_

## Referee Comment (RC1) · Anonymous Referee #1 · 26 May 2020

The topic is very important because little is known about the carbon balance of the Congo basin forests. However, the manuscript fails at bringing this missing information because of an unsuitable measurement technique and a very poor number of replicates (3 chambers only were used). Overall, the sampling strategy in poorly described; the site also. In addition, the manuscript is poorly written, and the discussion related to the stable isotope is too much speculative.

Page 1 A scientific article is different to a competitive grant proposal: let's try to stay humble. Remove "enormous" (line 3), "for the first time" (line 4) Line 6: Respiration in montane forest soils => Soil respiration in a montane forest. To avoid confusion, use

either soil respiration or soil CO2 efflux but avoid mixing the two. Line 8-10: be more precise. What are the differences that lead to this suggestion? And this suggestion is quite speculative because you compare soil respiration and soil C, but soil respiration also includes root and rhizopheric respiration that are less connected with the isotope composition of the soil C. It was only a suggestion line 10 but it becomes a firm conclusion line 14. This is annoying.

Page 2 Line 4: Ruehr 2010 is not the correct citation. In addition, there are lots of much older papers to cite here line 5: "respiration of organic matter" has no meaning. Soil organic matter is not a living organism. Overall, the paragraph in lines 3 to 9 is poorly written and lacks logical structure line 12-13: What is the reason why a high flux of CO2 and a high production would indicate a rapid turnover of C. Turnover (or mean residence time, the inverse) are flux divided by stock line 27: What do you mean by soil CO2 consumption What is the link between an increase in air temperature (line 31) and the length of the dry season (line 34). To my knowledge, the dry season in the Congo is cooler than the rainy season. At least, it does not indicate change in temperature (line 35) Page 3 Isotopic signatures of leaf-litter, soil organic carbon, soil-respired CO2 and dissolved stream water CO2 are not enough to determine sources of soil respired CO2. And what are the sinks? A bit wordy here. of soil respired CO2. Information on the stand structure are missing (at least tree density and basal area), as well as dominant species. Fine root biomass would also bring valuable information for comparing the two sites Page 4 line 3-4: The duration of the measurement is therefore not three years in contrast to what is claimed in the abstract line 7: three chambers for one site! That is definitely not enough to cover spatial variability. If there is something that is well documented in tropical forests, this is the large spatial variability. line 10: Static chambers installed for 12 hours! Does the chamber remain in the same place for two years? Measurements last one hour. Were the chambers opened before and after? 1 hour is already very long. How much CO2 accumulate during this time? Based on the keeling plots, concentration seems to vary from 500 to 2000ppm. It is far from the state of the art in terms of measurement of soil CO2 efflux.

page 5 line 10: this equation was used by many before Imer 2013 Line 31: how many litter traps? Line 33: how many soil samples?

page 11, line 31: Universally???? Page 12, line 13-16: it's totally speculative. Root respiration may contribute differently in the type of forest. Nothing allows the authors to test their hypothesis that decomposition is faster and soil microbes carbon-limited.

---

## Author Comment (AC1) · 31 May 2020

We thank the Referee for their review. We have identified three main issues raised by the referee and will respond to each below.

The static manual chamber method used in this study is a well-established method used to measure soil GHG fluxes (e.g. Garcia-Montiel et al., 2004; Imer et al., 2013; Werner et al., 2014; Courtois et al., 2018). While we acknowledge that more advanced methods to measure GHG fluxes exist (i.e. portable gas analyzers), logistical constraints of working at four remote sites in the Democratic Republic of the Congo for extended periods of time prohibited methods that required multiple expensive instru-

[Figure]

ments and reliable access to electricity. We chose the well-established methodology of evacuated gas sampling of static chambers because they 1.) were cost effective for sampling multiple sites for 2.5 years, 2.) were simple to perform for our field assistants, 3.) did not require electricity, and 4.) did not require materials that could be stolen or easily damaged. In conclusion, to conduct a long-term survey in the Congo Basin, we decided the exetainer sampling of permanently installed static chambers was the only reliable and suitable technique.

In response to the concern regarding the 1 hour sampling duration, none of the total 1108 individual soil $CO_2$ flux measurements showed saturation of $CO_2$ concentration in the chambers (i.e. by reaching a plateau, see examples for each site in Figure 1). All of the measured fluxes exhibited linear increases with very high $r^2$ (see Figure 2 below) and only fluxes with a $r^2 > 0.9$ were considered in our analyses. Furthermore, we would like to point out that, in the absence of chamber saturation effects, longer flux durations result in more accurate flux calculations, since the $\Delta CO_2$ is larger for each time interval relative to the measurement accuracy. Nevertheless, we thank the reviewer for requesting more detailed method descriptions and will edit the manuscript accordingly.

Regarding the number of replications, we used a minimum of three chambers per site as replicates which we described in the methods section. During the short-term sampling campaigns, five chambers were used. The results from these short-term campaigns with the additional chambers showed extremely low variability between chambers at the same site. We only conducted the long-term measurements in the montane forest with only three chambers in the same locations because the sampling material and time of our field assistants were limited given the logistics with doing research in the DRC. Moreover, three of our sites were located in the lowland forests (Yangambi, Djolu and Yoko) and separated by more than 100 km. These sites exhibited both low intra- and inter-site variability, further confirming that our number of replicates was sufficient for measuring soil $CO_2$ fluxes from these forests. We will add more detailed

information to the manuscript.

While we feel the inclusion of stable isotopes is relevant and of interest to the reader, we agree that certain points of the discussion are maybe somewhat speculative. We will tone down the language of this section in the revised draft and offer carbon limitation as merely a possible explanation for the observed trends.

In addition to the three main issues, several comments on word choice and phrasing were made. We thank the Referee for their suggestions and will amend the revised manuscript accordingly.

References:

Courtois, Elodie A.and Stahl, C., Van den Berge, J., Bréchet, L., Van Langenhove, L., Richter, A., Urbina, I., Soong, J. L., Peñuelas, J., and Janssens, I. A.: Spatial Variation of Soil $CO_2$, $CH_4$ and $N_2O$ Fluxes Across Topographical Positions in Tropical Forests of the Guiana Shield, Ecosystems, 22, 228–228, https://doi.org/10.1007/s10021-018-0281-x, 2018.

Garcia-Montiel, D. C., Melillo, J. M., Steudler, P. A., Tian, H., Neill, C., Kick-lighter, D.W., Feigl, B., Piccolo, M., and Cerri, C. C.: Emissions of $N_2O$ and $CO_2$ from terra firme forests in Rondonia, Brazil, Ecological Applications, 14, 214–220, https://doi.org/10.1890/01-6023, 2004.

Imer, D., Merbold, L., Eugster, W., and Buchmann, N.: Temporal and spatial variations of soil $CO_2$, $CH_4$ and $N_2O$ fluxes at three differently managed grasslands, Biogeosciences, 10, 5931–5945, https://doi.org/10.5194/bg-10-5931-2013, 2013.

Werner, C., Reiser, K., Dannenmann, M., Hutley, L. B., Jacobeit, J., Butterbach-Bahl, K.: $N_2O$, NO, $N_2$ and $CO_2$ emissions from tropical savanna and grassland of northern Australia: an incubation experiment with intact soil cores, Biogeosciences, 11, 6047–6065, https://doi.org/10.5194/bg-11-6047-2014, 2014.

[Figure]

**Fig. 1.** Examples of CO2 concentration over time during a single chamber measurement for each site

[Figure]

Fig. 2. Histogram of the r2 values for each linear fit

---

## Referee Comment (RC2) · Anonymous Referee #2 · 25 Jun 2020

General comments:

The paper by Baumgartner et al on soil CO2 emissions from tropical rainforests in the Congo Basin is relevant and mostly well written. It addresses the knowledge gap on GHG fluxes from the African continent, which is still critically under-researched and represents one of the main causes of uncertainty in global GHG budgets. The paper is generally well structured, and the results mostly support the drawn conclusions. There are a few areas, however, that could to with a little revision and rewriting, and some of the conclusions based on the isotopic signature of different ecosystem C compartments might be a little speculative and could do with some rephrasing. Furthermore,

some details on the experimental setup are missing and should be added to the materials and methods section. The statistics are sound but could be presented in a more attractive format. But if the authors address these concerns in an adequate manner, I am convinced that this paper can be a valuable contribution to Biogeosciences.

Specific comments:

Introduction p.1 L17: fungi are also considered to be microorganisms. Therefore it is enough to say microbial respiration, or alternatively, fungal and bacterial respiration. L19: the reference for global C flux via soil photosynthesis is a bit old, I suggest using the numbers from the latest IPCC report. p. 2 L15-34: The authors highlight why it is crucial to understand soil respiration especially in ecosystems that are less well researched (i.e. tropical African rainforests). This paragraph is a bit lengthy because I think the reader of Biogeosciences is aware of that fact. Please shorten this paragraph, and instead add some information on 13C partitioning throughout the C cascade of tropical rainforests, and what different d13C values can mean, as this will guide the reader towards the research questions. p. 3 L4-9: what were your hypotheses? Material and methods: p. 3 L15, L19-20, and throughout the manuscript: please don't confuse the terms "average" and "mean". The (geometric) mean is a form of the average, in addition to the median and the modus. It should, therefore, be "mean annual rainfall" and "mean annual temperature". This should also be addressed throughout the results and discussion section (e.g. mean flux, etc). p. 4: the section on soil $CO_2$ flux measurements lacks some important details: how big were the study areas and plots? How many plots were installed per site? What was the vegetation composition (dominant tree species, presence or absence of dense understorey, basal area of trees, etc)? Did you use 3 flux chambers per site or per plot (i.e. more per site)? How were the chambers arranged in plots (e.g. distance from large trees, understorey vegetation, depressions/mounds, etc)?

One more note on the number of replicates for $CO_2$ flux measurements: This is not 100% clear from the authors' description, but if I understand correctly, only 3 flux chambers were installed per study site. This is critical because spatial heterogeneity of soil respiration has been described in numerous previous studies, and this could lead to under- or overestimation of soil flux estimates. However, there are a couple of points that the authors could use to address this shortcoming: first, they have measured soil CO2 flux not only in one but in 3 lowland rainforests, and they could look at the difference between sites to describe spatial heterogeneity in the region. Second, if the flux chambers were always installed following a similar scheme, e.g. always at a fixed distance from trees, they would still be comparable even if not 100% representing absolute fluxes. Third, data on GHG fluxes from Africa are very scarce, and one of the reasons is the difficulty in getting research material into or out of the respective countries. I know from personal experience that it can be very difficult to buy or import even simple building material to construct flux chambers, and shipping of environmental samples can be complicated and often requires a lot of paperwork. I can imagine that the situation in DRC might have been similar. Therefore, for future studies on GHG fluxes in regions that are not easily accessible, I recommend the use of the gas-pooling technique by which gas samples from multiple (usually 3-6) chambers are put into the same GC vial, which can help to cover spatial heterogeneity while at the same time reducing the total number of samples. Nevertheless, even if the number of replicates is low and this probably introduces some uncertainty, this information on the magnitude of fluxes and their dynamics is still highly valuable, and I therefore still recommend the study for publication in BG.

p. 5 d13C measurements L24-25: wouldn't drawing 3 analytical samples of 20 ml each from the headspace of a 110 ml vial create an underpressure? How did the authors address this? L31: how many litter traps were installed per site, and how were they arranged? L33: how many soil samples were collected per site? What was their arrangement (e.g. distance to chambers, distance to trees, etc)? Statistics L6: you assumed little year-to-year variability of your data, but did you actually check if the climatic conditions (rainfall, temperature, moisture) varied between years? p. 6 Figure 2: there are two dips in WFPS in March and October in the lowland forest, where

WFPS dropped rapidly from c. 30 to 20%, and then recovered within a week or so – do you have an explanation for this? p. 7 Table 1: I know that the R output of lmer looks like this, but it's not very convenient for the reader to understand the results of the statistical analysis. For example, for d13C there is a significant effect of "Montane forest – stream CO2". This is ambiguous: does it mean that the d13C of stream-CO2 is different in montane and lowland forests? Or that the d13C of stream-CO2 is different from the d13C of the other compartments (soil CO2, litterfall, SOC) only in the montane but not the lowland forest? Please use a different way to present these results as they are critical. For example, as a start you could add letters/starts to Figure 2, presenting sign. differences between compartments via different letters, and differences between forests via stars (or something like that). L4-6: You state that stream-CO2 was significantly depleted in the wet season in lowland forests but not montane forests. However, in Table 1 you state "montane forest – wet season – stream CO2" to be significant. Isn't this contradictory (or just another example of how Table 1 could be misinterpreted)? Discussion p. 8 L5-7: Move this to the results section. p. 10 L9-12: Careful, while it is true that with increasing dry season length soil CO2 fluxes might decrease, but it is not clear how future more erratic rainfall patterns and the corresponding more extreme drying-rewetting events will affect respiration, and whether potential CO2 pulses after rewetting compensate or outweigh reduced soil respiration. L25-30: Good call! I agree that the correlations between soil CO2 flux and temperature in tropical systems that show very little annual variation should be handled with care. In your case, they might be significant simply because your sample size is large enough, but I would not over-interpret them. As you correctly state, moisture and C availability are likely the bigger players here. p.11 L1-4: soil moisture not only controls O2 diffusion but also the diffusion of C substrates to soil microorganisms. Therefore, the response of respiration to moisture is more often an effect of C limitation (at low moisture) than an effect of O2 limitation (which really only becomes critical at very high moisture contents). Please add this to the discussion, and I recommend these papers on the mechanisms underlying this: Manzoni S, Moyano F, Kätterer T, Schimel J (2016)

Modeling coupled enzymatic and solute transport controls on decomposition in drying soils. Soil Biol Biochem 95:275–287. doi: 10.1016/j.soilbio.2016.01.006 Moyano FE, Manzoni S, Chenu C (2013) Responses of soil heterotrophic respiration to moisture availability: An exploration of processes and models. Soil Biol Biochem 59:72–85. doi: 10.1016/j.soilbio.2013.01.002 Moyano FE, Vasilyeva N, Bouckaert L, et al (2012) The moisture response of soil heterotrophic respiration: Interaction with soil properties. Biogeosciences 9:1173–1182. doi: 10.5194/bg-9-1173-2012 L20: you mention photosynthesis, yet this was not measured and is therefore a bit speculative. p. 12 L4: which canopy processes other than photosynthesis could those be? Furthermore, how do you think that vegetation composition might affect d13C, and could this explain differences between lowland and montane forests? Can different trees have different leaf d13C signatures, which could be reflected throughout the C cascade? L6: what are the mechanisms underlying the enrichment of 13C at lower temperatures? Conclusions This is mostly a repetition of the results. Please instead give the "message of the story" – what are the implications of the results you found? What are questions that remain open? And what have we learned? L24: how were the sites different in vegetation composition? Please describe in the M&M section and also address in discussion L27: what does this indicate, that that there was no temperature dependency of soil respiration between sites? p. 13 L4: you conclude the paper with the statement that these forests might become C sources under a warming climate, yet you did not find a strong effect of temperature! Instead, you could state that changes of C balance might happen in response to more erratic rainfalls and weather extremes. Appendix A: Method supplement L6-25: please use the past tense throughout this section. p. 16 Figure 3A: change x-axis labels of panel d to the format HH:MM (e.g. 10:00, 15:00, . . .) to make it clear that those are hours. Technical corrections: p. 5 L30: please correct ". . .during the wet season from October to May" p. 6 L9: please correct "values were found" (use past tense throughout the results section) L8 and elsewhere: You very often use the term "respectively"; however, I'm not a big fan of it, for two reasons: first, sentences become very complex and sometimes hard to understand when using this

term, and second, it forces the reader to jump back and forth between the end and the start of the sentence, which disrupts the flow of reading. Very often, you'll find that your sentence won't actually become any longer if instead of using "respectively", you describe the results one after the other, in this case, this could be "The mean [instead of "average", see my earlier comment] annual values we measured in this study in the Congo basin, which are 3.83 $\mu$mol m-2 s-1 for the montane forest and 3.69 $\mu$mol m-2 s-1 for the lowland forest, are within the range of reported values from other tropical forests." I propose that you revise the MS and try to reduce the use of "respectively". This will make the paper easier to follow. L14: please rearrange "... and they were rather low compared to our flux rates" p.9 L2: "...showed marked seasonality [comma] with a 34 % decrease during the dry season [comma] whereas..." L4: please rephrase "however, the decrease they found was not as pronounced as..." p. 10 L22: "statistically significant correlation" L18: please rephrase "play a crucial role in controlling soil respiration" p. 11: L4: please add a comma here, otherwise the phrase is misleading: "stress soil microbial communities, and autotrophic respiration" L10: please rephrase as this is otherwise misleading "While soil respiration in lowland forests is most likely C-limited, respiration in montane forests seems to be more sensitive to environmental conditions and could represent a potentially large C source with climate change." p. 12 L17: enrichment does not occur in the "location" but in the movement from one compartment to another. Please rephrase "the highest enrichment occurs in the last step from soil to stream-dissolved $CO_2$". L25: please rephrase: "However, in contrast to the lowland forest, the montane forest site exhibited strong seasonality of soil respiration, primarily driven by WFPS during the dry season."

---

## Author Comment (AC2) · 10 Jul 2020

Received and published 25 June 2020.

General comments:

The paper by Baumgartner et al on soil CO2 emissions from tropical rainforests in the Congo Basin is relevant and mostly well written. It addresses the knowledge gap on GHG fluxes from the African continent, which is still critically under-researched and represents one of the main causes of uncertainty in global GHG budgets. The paper is generally well structured, and the results mostly support the drawn conclusions. There are a few areas, however, that could to with a little revision and rewriting, and some of the conclusions based on the isotopic signature of different ecosystem C compartments might be a little speculative and could do with some rephrasing. Furthermore, some details on the experimental setup are missing and should be added to the materials and methods section. The statistics are sound but could be presented in a more attractive format. But if the authors address these concerns in an adequate manner, I am convinced that this paper can be a valuable contribution to Biogeosciences.

We thank the Referee for the positive review and constructive comments. We take their point that some conclusions based on the isotopic signatures are somewhat speculative and we will tone them down in the revised manuscript. We will also edit the materials and methods section so that the experimental set up is clearer to the readers. Furthermore, we will remake Figure 3 and include information from the statistical model. To avoid confusion, we will split up Table 1 into two parallel analyses (one for montane and one for lowland).

Specific comments:
Introduction p.1 L17: fungi are also considered to be microorganisms. Therefore it is enough to say microbial respiration, or alternatively, fungal and bacterial respiration.

Thank you for the specificity, this will be changed in the manuscript.

L19: the reference for global C flux via soil photosynthesis is a bit old, I suggest using the numbers from the latest IPCC report.

In this sentence, we present the global soil respiration rate from Bond-Lamperty and Thomson, 2010. To our knowledge, in the latest IPCC report, soil respiration rates were only presented together with the C loss via fires.

p. 2 L15-34: The authors highlight why it is crucial to understand soil respiration especially in ecosystems that are less well researched (i.e. tropical African rainforests). This paragraph is a bit lengthy because I think the reader of Biogeosciences is aware of that fact. Please shorten this paragraph, and instead add some information on 13C partitioning throughout the C cascade of tropical rainforests, and what different d13C values can mean, as this will guide the reader towards the research questions.

We agree that we can condense this paragraph and introduce the [13]C partitioning.

p. 3 L4-9: what were your hypotheses?

Since our objectives were to quantify annual soil $CO_2$ fluxes from forests of the Congo Basin and to assess differences between forest types, this was primarily a descriptive study. As such, we did not formulate specific hypotheses to test. Furthermore, it would not be correct scientific practice to formulate hypotheses after the fact. Hence, we refrained from formulating hypotheses now.

Material and methods: p. 3 L15, L19-20, and throughout the manuscript: please don't confuse the terms "average" and "mean". The (geometric) mean is a form of the average, in addition to the median and the modus. It should, therefore, be "mean annual rainfall" and "mean annual temperature". This should also be addressed throughout the results and discussion section (e.g. mean flux, etc).

This will be changed in the manuscript.

p. 4: the section on soil CO2 flux measurements lacks some important details: how big were the study areas and plots? How many plots were installed per site? What was the vegetation composition (dominant tree species, presence or absence of dense understorey, basal area of trees, etc)? Did you use 3 flux chambers per site or per plot (i.e. more per site)? How were the chambers arranged in plots (e.g. distance from large trees, understorey vegetation, depressions/mounds, etc)?

We thank the reviewer for requesting more details regarding the flux measurements. We installed one plot in a mixed forest site in the montane forest. There we used seven chambers for the short-term campaigns and three chambers in the first year of long-term measurements. In the second year of the long-term measurements, we increased the number of chambers to five. In the lowland forests of Yoko and Yangambi, we installed two plots in each site, one in a mixed forest and one in a mono-dominant forest, where more than 60% of the basal area consists of the species *Gilbertiodendron dewevrei.* For the short-term campaigns, we used four chambers in the mixed forest and three chambers in the mono-dominant forest. We started the long-term campaign with four chambers in the mixed forest and two chambers in the mono-dominant forest and after a year, we proceeded with five chambers in the mixed forest and stopped sampling in the mono-dominant forest. The chambers were randomly placed between trees and we avoided hills and depressions. We will make the respective section of the manuscript clearer and add more information on the study areas (dominant tree species, understory presence, basal area of trees, etc.) and chamber arrangements.

One more note on the number of replicates for CO2 flux measurements: This is not 100% clear from the authors' description, but if I understand correctly, only 3 flux chambers were installed per study site. This is critical because spatial heterogeneity of soil respiration has been described in numerous previous studies, and this could lead to under- or overestimation of soil flux estimates. However, there are a couple of points that the authors could use to address this shortcoming: first, they have measured soil CO2 flux not only in one but in 3 lowland rainforests, and they could look at the difference between sites to describe spatial heterogeneity in the region. Second, if the flux chambers were always installed following a similar scheme, e.g. always at a fixed distance from trees, they would still be comparable even if not 100% representing absolute fluxes. Third, data on GHG fluxes from Africa are very scarce, and one of the reasons is the difficulty in getting research material into or out of the respective countries. I know from personal experience that it can be very difficult to buy or import even simple building material to construct flux chambers, and shipping of environmental samples can be complicated and often requires a lot of paperwork. I can imagine that the situation in DRC might have been similar. Therefore, for future studies on GHG fluxes in regions that are not easily accessible, I recommend the use of the gas-pooling technique by which gas samples from multiple (usually 3-6) chambers are put into the same GC vial, which can help to cover spatial

heterogeneity while at the same time reducing the total number of samples. Nevertheless, even if the number of replicates is low and this probably introduces some uncertainty, this information on the magnitude of fluxes and their dynamics is still highly valuable, and I therefore still recommend the study for publication in BG.

Thank you for this comment. As stated above, we used a minimum of three chambers per site. However, for the short-term campaigns, we used up to seven chambers per site. The results from these short-term campaigns showed relatively low variability between chambers at the same site (C.V. of 22% in the lowland forest and 16% in the montane forest) and thus we decided to reduce the number of chambers due to the limited number of evacuated vials for gas sampling (four were used per chamber per sampling for this study). Additionally, the reviewer correctly acknowledged that we measured $CO_2$ fluxes at three different lowland forest sites, which are separated by more than 100 km. The average fluxes of these three sites were also similar (inter-site C.V. of 25%) which gave us further confidence that even those periods where only three chambers were used are representative. We will add this information to the manuscript. As the reviewer points out already, it was not easy to get materials into the DRC. Therefore, we were only able to increase replication with additional visits to the sites, taking more material to the DRC. Ideally, this would have been done from the start but was not possible due to logistical constraints. Moreover, we appreciate the suggestion to use gas-pooling and will consider adopting this technique in the future.

p. 5 d13C measurements L24-25: wouldn't drawing 3 analytical samples of 20 ml each from the headspace of a 110 ml vial create an underpressure? How did the authors address this?

When withdrawing the samples from the 110 mL vials, a luer-stopcock between syringe and needle was used to avoid underpressure problems when removing the needle from the vial headspace during subsampling. That is, after withdrawal of 25 mL of sample, the luer-stopcock valve between needle and syringe was closed and the syringe was removed from the headspace. After, the plunger was pushed to 20 mL before opening the valve and injecting the subsample to 20 mL Labco vials. This procedure was repeated 3 times. The precision of three analytical replicates was excellent, with a maximum standard deviation of 0.25‰. We will add a more detailed description of the sampling to the manuscript.

L31: how many litter traps were installed per site, and how were they arranged? L33: how many soil samples were collected per site? What was their arrangement (e.g. distance to chambers, distance to trees, etc)?

Eight litter traps were installed per site and arranged in two rows of four. There was a distance of eight meters between traps. Soil samples were collected at three random positions at each site. We thank the reviewer for their attention to detail and will add this information to the manuscript.

Statistics L6: you assumed little year-to-year variability of your data, but did you actually check if the climatic conditions (rainfall, temperature, moisture) varied between years?

We compiled the flux, temperature and WFPS data in weekly bins for easier presentation of the seasonality of the fluxes. However, for statistical analysis (influence of soil temperature and soil moisture on soil $CO_2$ fluxes) we used the individual fluxes with the actual soil temperature and soil moisture conditions during each flux measurement. As a result, the weekly bins did not affect the results of the statistical analysis.

p. 6 Figure 2: there are two dips in WFPS in March and October in the lowland forest, where WFPS dropped rapidly from c. 30 to 20%, and then recovered within a week or so – do you have an explanation for this?

The two observed dips are located right within the peak rainy season, therefore, speedy recovery of soil moisture can be expected.

p. 7 Table 1: I know that the R output of lmer looks like this, but it's not very convenient for the reader to understand the results of the statistical analysis. For example, for d13C there is a significant effect of "Montane forest – stream CO2". This is ambiguous: does it mean that the d13C of stream-CO2 is different in montane and lowland forests? Or that the d13C of stream- CO2 is different from the d13C of the other compartments (soil CO2, litterfall, SOC) only in the montane but not the lowland forest? Please use a different way to present these results as they are critical. For example, as a start you could add letters/starts to Figure 2, presenting sign. differences between compartments via different letters, and differences between forests via stars (or something like that).

We agree with the reviewer that we can improve this presentation, to ensure unambiguous interpretation. The p-values (as well as $R^2$s) in the case of linear mixed effect models are only estimations of p-values, and should be interpreted with caution either way, hence the interpretation of the table was mainly meant for the effect sizes. We suggest that we remake Figure 3 and include the model information in a new Figure 3. For the ease of interpretation, we will split both the figure and modeling up in two parallel analyses (one for lowland, one for montane). This will avoid confusion with the interpretation of too many interaction effects, but the effect sizes of both models will still allow the reader to interpret both inter and intra-forest type effects on $\delta^{13}C$. A table of this 'split-up' analysis (much like the current Table 1), could then go to supplementary materials. We thank the reviewer for making this clear, it is in our best interest that the readership of the paper can easily interpret the data we show.

L4-6: You state that stream-CO2 was significantly depleted in the wet season in lowland forests but not montane forests. However, in Table 1 you state "montane forest – wet season – stream CO2" to be significant. Isn't this contradictory (or just another example of how Table 1 could be misinterpreted)?

In Table 1 "Montane forest – Wet season – Stream CO2" has a P-value of 0.42. However, we fully agree that the current table, including three factors and two interaction effects, creates confusion, and will change this for the next version of this manuscript.

Discussion p. 8 L5-7: Move this to the results section.

Thank you for the suggestion, this will be moved to the suggested section in the revised draft.

p. 10 L9-12: Careful, while it is true that with increasing dry season length soil CO2 fluxes might decrease, but it is not clear how future more erratic rainfall patterns and the corresponding more extreme drying-rewetting events will affect respiration, and whether potential CO2 pulses after rewetting compensate or outweigh reduced soil respiration.

We thank the reviewer for providing this qualification and will rewrite this statement to reflect a larger degree of uncertainty, for example, that there could also be $CO_2$ pulses that compensate for lower respiration as a result of these extreme drying-rewetting events.

L25-30: Good call! I agree that the correlations between soil CO2 flux and temperature in tropical systems that show very little annual variation should be handled with care. In your case, they might

be significant simply because your sample size is large enough, but I would not over-interpret them. As you correctly state, moisture and C availability are likely the bigger players here.

We are glad that the reviewer agrees with our conservative interpretation.

p.11 L1-4: soil moisture not only controls $O_2$ diffusion but also the diffusion of C substrates to soil microorganisms. Therefore, the response of respiration to moisture is more often an effect of C limitation (at low moisture) than an effect of $O_2$ limitation (which really only becomes critical at very high moisture contents). Please add this to the discussion, and I recommend these papers on the mechanisms underlying this: Manzoni S, Moyano F, Kätterer T, Schimel J (2016) Modeling coupled enzymatic and solute transport controls on ecomposition in drying soils. Soil Biol Biochem 95:275–287. doi: 10.1016/j.soilbio.2016.01.006 Moyano FE, Manzoni S, Chenu C (2013) Responses of soil heterotrophic respiration to moisture availability: An exploration of processes and models. Soil Biol Biochem 59:72–85. doi: 10.1016/j.soilbio.2013.01.002 Moyano FE, Vasilyeva N, Bouckaert L, et al (2012) The moisture response of soil heterotrophic respiration: Interaction with soil properties. Biogeosciences 9:1173–1182. doi: 10.5194/bg-9-1173-2012

We thank the reviewer for this nuanced perspective and will integrate the point along with the mentioned references into the discussion.

L20: you mention photosynthesis, yet this was not measured and is therefore a bit speculative.

We agree with the reviewer that using the term "photosynthesis" here is a bit misleading in the sense that it indeed does sound this parameter had been measured. We suggest to re-write this section avoiding the term photosynthesis: "In this study, the link between C assimilation and soil $CO_2$ is evident through […]"

p. 12 L4: which canopy processes other than photosynthesis could those be? Furthermore, how do you think that vegetation composition might affect d13C, and could this explain differences between lowland and montane forests? Can different trees have different leaf d13C signatures, which could be reflected throughout the C cascade?

These are two very good points raised by the reviewer. We were mainly thinking of stomatal conductance as the other important canopy process determining [13]C discrimination. Since there is an interplay between photosynthesis and stomatal conductance on [13]C discrimination, we lumped these two processes together (as canopy processes). We will specify in the text how we define canopy processes.

The effect of altitude on $\delta^{13}C$ of canopy leaves is well known (Körner et al., 1988, Hultine & Marshall, 2000; Chen et al., 2015) and can be explained by a combination of factors and the two consistent patterns associated with increasing elevation are a decrease in atmospheric pressure and in temperature. The decrease in $O_2$ partial pressure and temperature supposedly promotes a decline in ci/ca and the direct implication of this decline is that $\delta^{13}C$ values become less negative (Wang et al., 2017). We will further elaborate on this issue in the new version of the manuscript.

L6: what are the mechanisms underlying the enrichment of 13C at lower temperatures?

In the subsequent sentence, we explain that temperature changes can result in shifts in microbial communities, which can impact fractionation during heterotrophic soil respiration (Andrews et al., 2000).

Conclusions This is mostly a repetition of the results. Please instead give the "message of the story" – what are the implications of the results you found? What are questions that remain open? And what have we learned?

We will rephrase the conclusion and add implications of our results and address remaining research questions.

L24: how were the sites different in vegetation composition? Please describe in the M&M section and also address in discussion

We will add this to the site description and discuss it accordingly.

L27: what does this indicate, that that there was no temperature dependency of soil respiration between sites?

As our results suggests that respiration in the lowland forests is substrate limited, we reason that the higher temperatures, compared to the montane forest, will not results in an increased soil $CO_2$ flux. We amend tis sentence and clarify this in the manuscript.

p. 13 L4: you conclude the paper with the statement that these forests might become C sources under a warming climate, yet you did not find a strong effect of temperature! Instead, you could state that changes of C balance might happen in response to more erratic rainfalls and weather extremes.

We will rephrase the final statement in our conclusion in a manner that it better fits our observations.

Appendix A: Method supplement L6-25: please use the past tense throughout this section.

The Method supplement will be modified to the past tense.

p. 16 Figure 3A: change x-axis labels of panel d to the format HH:MM (e.g. 10:00, 15:00,...) to make it clear that those are hours.

The figure will be adjusted accordingly.

Technical corrections: p. 5 L30: please correct "...during the wet season from October to May"

This will be changed.

p. 6 L9: please correct "values were found" (use past tense throughout the results section)

The results section will be modified to the past tense.

L8 and elsewhere: You very often use the term "respectively"; however, I'm not a big fan of it, for two reasons: first, sentences become very complex and sometimes hard to understand when using this term, and second, it forces the reader to jump back and forth between the end and the start of the sentence, which disrupts the flow of reading. Very often, you'll find that your sentence won't actually become any longer if instead of using "respectively", you describe the results one after the other, in this case, this could be "The mean [instead of "average", see my earlier comment] annual values we measured in this study in the Congo basin, which are 3.83 _mol m-2 s-1 for the montane forest and 3.69 _mol m-2 s-1 for the lowland forest, are within the range of reported values from

other tropical forests." I propose that you revise the MS and try to reduce the use of "respectively". This will make the paper easier to follow.

*We very much agree with this comment and will rephrase those sentences and try to make it easier to read.*

L14: please rearrange "…and they were rather low compared to our flux rates"

*The sentence will be rearranged.*

p.9 L2: "…showed marked seasonality [comma] with a 34 % decrease during the dry season [comma] whereas.."

*We will add the commas.*

L4: please rephrase "however, the decrease they found was not as pronounced as…"
p. 10 L22: "statistically significant correlation"
L18: please rephrase "play a crucial role in controlling soil respiration"
p. 11: L4: please add a comma here, otherwise the phrase is misleading: "stress soil microbial communities, and autotrophic respiration"
L10: please rephrase as this is otherwise misleading "While soil respiration in lowland forests is most likely C-limited, respiration in montane forests seems to be more sensitive to environmental conditions and could represent a potentially large C source with climate change."
p. 12 L17: enrichment does not occur in the "location" but in the movement from one compartment to another. Please rephrase "the highest enrichment occurs in the last step from soil to stream-dissolved CO2".
L25: please rephrase: "However, in contrast to the lowland forest, the montane forest site exhibited strong seasonality of soil respiration, primarily driven by WFPS during the dry season."

*We will rephrase these sentences and will add commas where needed.*

**References**

Andrews, J. A., Matamala, R., Westover, K. M., and Schlesinger, W. H.: Temperature effects on the diversity of soil heterotrophs and the δ13C of soil-respired CO2, Soil Biology and Biochemistry, 32, 699 – 706, 2000.

Chen, L., Flynn, D. F. B., Zhang, X., Gao, X., Lin, L., Luo, J., Zhao, C.: Divergent patterns of foliar δ13C and δ15N in Quercus aquifolioides with an altitudinal transect on the Tibetan Plateau: an integrated study based on multiple key leaf functional traits, Journal of Plant Ecology, 8, 303 – 312, 2015.

Hultine, K. R., Marshall, J. D.: Altitude trends in conifer leaf morphology and stable carbon isotope composition. Oecologia, 123, 32 – 40, 2000.

Körner, C., Farquhar, G. D., Roksandic, Z.: A global survey of carbon isotope discrimination in plants from high altitude. Oecologia, 74, 623 – 632, 1988.

Wang, M., Liu, G., Jin, T., Li, Z., Gong, L., Wang, H., Ye, X.: Age- related changes of leaf traits and stoichiometry in alpine shrub (Rhododendron agglutinatum) along altitudinal gradient. Journal of Mountain Science, 14, 106-118, 2017.

---

## Author Response (AR1)

**Response to Reviewers and Editor**

We thank the editor and both reviewers for the constructive and thorough reviews of our manuscript. Here we present our responses to reviewer comments and the revised manuscript. We sincerely hope the changes satisfy both reviewers. We believe the quality of the manuscript has improved substantially thanks to these reviews. We also improved the calculations of the $CO_2$ fluxes and changed the values in the manuscript. To keep it clear for both the editor and the reviewers, we put the original comments in grey, our response in black.

**Anonymous Referee #1**

The topic is very important because little is known about the carbon balance of the Congo basin forests. However, the manuscript fails at bringing this missing information because of an unsuitable measurement technique and a very poor number of replicates (3 chambers only were used). Overall, the sampling strategy in poorly described; the site also. In addition, the manuscript is poorly written, and the discussion related to the stable isotope is too much speculative.

We thank the Referee for their review. We acknowledge that the referee is critical and identified three main issues raised by the referee. We will respond to each of these issues below and present our changes in the manuscript.

Page 1 A scientific article is different to a competitive grant proposal: let's try to stay humble. Remove "enormous" (line 3), "for the first time" (line 4)

We removed "enormous" and "for the first time" from the manuscript (p.1, L. 2-3)

Line 6: Respiration in montane forest soils => Soil respiration in a montane forest. To avoid confusion, use either soil respiration or soil CO2 efflux but avoid mixing the two.

We thank the reviewer for this observation and we amended the manuscript where necessary to get a more coherent text throughout the manuscript.

Line 8-10: be more precise. What are the differences that lead to this suggestion? And this suggestion is quite speculative because you compare soil respiration and soil C, but soil respiration also includes root and rhizopheric respiration that are less connected with the isotope composition of the soil C. It was only a suggestion line 10 but it becomes a firm conclusion line 14. This is annoying.

We made the sentence in Line 9 clearer and toned down the conclusion in line 14.

Page 2 Line 4: Ruehr 2010 is not the correct citation. In addition, there are lots of much older papers to cite here

The reviewer is right. We corrected the citation to "Ruehr et al., 2010", and added "Rustad et al., 2000" (p. 2, L. 4).

line 5: "respiration of organic matter" has no meaning. Soil organic matter is not a living organism.

We adjusted the sentence to: "… whereas soil moisture affects the diffusion of C substrate, atmospheric oxygen and respired $CO_2$ through soil pores." (p.2, L. 5).

Overall, the paragraph in lines 3 to 9 is poorly written and lacks logical structure

This paragraph has been edited now by a native speaker (p.2, L. 3-9).

line 12-13: What is the reason why a high flux of CO2 and a high production would indicate a rapid turnover of C. Turnover (or mean residence time, the inverse) are flux divided by stock

We changed this sentence and avoided the term "C turnover" (p. 2, L. 13 - 15).

line 27: What do you mean by soil CO2 consumption

By the term "$CO_2$ consumption" we meant photosynthesis, however, we removed the term from the manuscript as it does not fit (p. 2, L 28).

What is the link between an increase in air temperature (line 31) and the length of the dry season (line 34). To my knowledge, the dry season in the Congo is cooler than the rainy season. At least, it does not indicate change in temperature (line 35)

We thank the reviewer for identifying this point of confusion. In this paragraph, we want to emphasize that changes in temperature and precipitation affect soil respiration. We have now adjusted the whole paragraph to make this clear (p. 2, L 30 to 32).

Page 3 Isotopic signatures of leaf-litter, soil organic carbon, soil respired CO2 and dissolved stream water CO2 are not enough to determine sources of soil respired CO2. And what are the sinks? A bit wordy here. of soil respired CO2.

We rephrased this sentence (p. 3, L. 21).

Information on the stand structure are missing (at least tree density and basal area), as well as dominant species. Fine root biomass would also bring valuable information for comparing the two sites

We thank the reviewer for identifying these parameters as important missing data in the manuscript. We have now added information on basal area and dominant species where it was available (p. 3, L. 28-29;  p.4, 1-4). Unfortunately, we do not have information on fine root biomass.

Page 4 line 3-4: The duration of the measurement is therefore not three years in contrast to what is claimed in the abstract

We thank the reviewer for pointing this out. In the abstract we meant measurements during three calendar years, we adjusted the wording in the abstract (p. 1, L 5).

line 7: three chambers for one site! That is definitely not enough to cover spatial variability. If there is something that is well documented in tropical forests, this is the large spatial variability.

Regarding the number of replications, we used a minimum of three chambers per site as replicates which we described in the methods section. During the short-term sampling campaigns, seven chambers were used. The results from these short-term campaigns with the additional chambers showed extremely low variability between chambers at the same site. We only conducted the first year of the long-term measurements in the montane forest with only three chambers in the same locations because the sampling material and time of our field assistants were limited given the logistics with doing research in the DRC. Moreover, three of our sites were located in the lowland forests (Yangambi, Djolu and Yoko) and separated by more than 100 km. These sites exhibited both low intra- and inter-site variability, further confirming that our number of replicates was sufficient for measuring soil $CO_2$ fluxes from these forests. We have added more information about the numbers

of chambers used in the subsection "Soil $CO_2$ flux measurements" of the Methods section (p. 5, L. 7-14) and present CV values for inter- and intra-site variability in the result section (p. 8, L. 6-9).

line 10: Static chambers installed for 12 hours! Does the chamber remain in the same place for two years? Measurements last one hour. Were the chambers opened before and after? 1 hour is already very long. How much CO2 accumulate during this time? Based on the keeling plots, concentration seems to vary from 500 to 2000ppm. It is far from the state of the art in terms of measurement of soil CO2 efflux.

The static manual chamber method used in this study is a well-established method used to measure soil GHG fluxes (e.g. Garcia-Montiel et al., 2004; Imer et al., 2013; Werner et al., 2014; Courtois et al., 2018). While we acknowledge that more advanced methods to measure GHG fluxes exist (i.e. portable gas analyzers, automated chambers, etc.), logistical constraints of working at four remote sites in the Democratic Republic of the Congo for extended periods of time prohibited methods that required multiple expensive instruments and reliable access to electricity. We chose the well-established methodology of evacuated gas sampling of static chambers because they 1.) were cost effective for sampling multiple sites for 2.5 years, 2.) were simple to perform for our field assistants, 3.) did not require electricity, and 4.) did not require materials that could be stolen or easily damaged. In addition to conducting scientific research, our group is interested in teaching and building capacity within our local student collaborators. The static chamber method, as opposed to automated chambers, allowed our students to play more significant role in the research. In conclusion, to conduct a long-term survey in the Congo Basin, we decided the exetainer sampling of permanently installed static chambers was the most suitable technique for our particular study.

Regarding the concerns over the chamber installation and measurement procedure. The chambers (PVC collars without lid) remained installed after initial placement. We waited with the first measurements for at least 12h after initial placement. The chambers were only closed for the duration of measurement (1h). In response to the concern regarding the 1 hour sampling duration, none of the total 1108 individual soil $CO_2$ flux measurements showed saturation of $CO_2$ concentration in the chambers (i.e. by reaching a plateau, see examples for each site in Figure 1). All of the measured fluxes exhibited linear increases with very high $r^2$ (see Figure 2) and only fluxes with a $r^2 >$ 0.9 were considered in our analyses. Furthermore, we would like to point out that, in the absence of chamber saturation effects, longer flux durations result in more accurate flux calculations, since the $\Delta CO_2$ is larger for each time interval relative to the measurement accuracy. Nevertheless, we thank the reviewer for requesting more detailed method descriptions and we edited the manuscript accordingly (p. 5, L. 19, 24-25).

[Figure]

*Figure 1: Examples of CO2 concentration over time during a single chamber measurement for each site*

[Figure]

*Figure 2: Histogram of the $r^2$ values for each linear fit*

page 5 line 10: this equation was used by many before Imer 2013

We thank the reviewer for pointing out that the equation was used prior to Imer 2013 and have replaced this reference with "Hutchinson and Mosier (1981)", which is, to our knowledge, the earliest usage (p. 6, L. 1).

Line 31: how many litter traps? Line 33: how many soil samples?

Eight litter traps were installed per site and arranged in two rows of four. There was a distance of eight meters between traps. Soil samples were collected at three random positions at each site. We thank the reviewer noting this unclarity and we now added this information to the manuscript (p. 6, L. 27-30).

page 11, line 31: Universally????

Indeed, this was too strong of an adverb to use. We replaced the word "universally" with "generally" (p.13, L. 11).

Page 12, line 13-16: it's totally speculative. Root respiration may contribute differently in the type of forest. Nothing allows the authors to test their hypothesis that decomposition is faster and soil microbes carbon-limited.

While we feel the inclusion of stable isotopes is relevant and of interest to the reader, we agree that certain points of the discussion are maybe somewhat speculative. We have toned down the language of this section in the revised draft and offer carbon limitation as merely a possible explanation for the observed trends (p. 13, L. 33-35). Moreover, we offer the caveat that contributions of $CO_2$ from root respiration can vary with forest type, which may confound inter-site comparison (p. 14, L. 1-3).

Anonymous Referee #2

General comments:
The paper by Baumgartner et al on soil CO2 emissions from tropical rainforests in the Congo Basin is relevant and mostly well written. It addresses the knowledge gap on GHG fluxes from the African continent, which is still critically under-researched and represents one of the main causes of uncertainty in global GHG budgets. The paper is generally well structured, and the results mostly support the drawn conclusions. There are a few areas, however, that could to with a little revision and rewriting, and some of the conclusions based on the isotopic signature of different ecosystem C compartments might be a little speculative and could do with some rephrasing. Furthermore, some details on the experimental setup are missing and should be added to the materials and methods section. The statistics are sound but could be presented in a more attractive format. But if the authors address these concerns in an adequate manner, I am convinced that this paper can be a valuable contribution to Biogeosciences.

We thank the Referee for the positive review and constructive comments. We take their point that some conclusions based on the isotopic signatures are somewhat speculative and have now toned them down in the revised manuscript. We added a line into the discussion that our interpretation is speculative and that further research is needed to test our hypothesis (p. 14, L. 2-3).

We also edited the materials and methods section so that the experimental set up is clearer to the readers (p. 5, L. 7-14 and p. 6, L. 27-30).

To avoid confusion, we split up the analysis of the $CO_2$ fluxes in two models: one for the lowland and one for the montane forest. Both results are now presented in a new Table 1 (p. 8). We also split up

the statistical analysis of the $\delta^{13}$C values into the two forest sites and present the values and significances in a new Figure 3 (p. 10). The seasonality of the $\delta^{13}$C values are now presented in a new supplementary figure (Figure A4, p. 19).

Specific comments:
Introduction p.1 L17: fungi are also considered to be microorganisms. Therefore it is enough to say microbial respiration, or alternatively, fungal and bacterial respiration.

Thank you for the specificity, we changed it to "… bacterial and fungal respiration" (p. 1, L. 17).

L19: the reference for global C flux via soil photosynthesis is a bit old, I suggest using the numbers from the latest IPCC report.

Unfortunately, in the latest IPCC report, soil respiration rates were only presented together with the C loss via fires. In order to report more recent numbers, as the reviewer suggests, we decided to use the numbers of the most recent global carbon project report (p. 1, L. 19).

p. 2 L15-34: The authors highlight why it is crucial to understand soil respiration especially in ecosystems that are less well researched (i.e. tropical African rainforests). This paragraph is a bit lengthy because I think the reader of Biogeosciences is aware of that fact. Please shorten this paragraph, and instead add some information on 13C partitioning throughout the C cascade of tropical rainforests, and what different d13C values can mean, as this will guide the reader towards the research questions.

We agree that the ideas can be condensed and have now shorted this paragraph. Additionally, we moved the beginning of the paragraph 4.4 to the introduction, to give some information about $^{13}$C partitioning (p.3, L. 5-14).

p. 3 L4-9: what were your hypotheses?

Since our objectives were to quantify annual soil $CO_2$ fluxes from forests of the Congo Basin and to assess differences between forest types, this was primarily a descriptive study. However, we added now that we expected higher soil $CO_2$ fluxes in the lowland forest compared to the montane forest, as we expected higher soil temperature and WFPS conditions (p. 1, L. 17-19) .

Material and methods: p. 3 L15, L19-20, and throughout the manuscript: please don't confuse the terms "average" and "mean". The (geometric) mean is a form of the average, in addition to the median and the modus. It should, therefore, be "mean annual rainfall" and "mean annual temperature". This should also be addressed throughout the results and discussion section (e.g. mean flux, etc).

We replaced the term "average" with "mean" throughout the manuscript.

p. 4: the section on soil CO2 flux measurements lacks some important details: how big were the study areas and plots? How many plots were installed per site? What was the vegetation composition (dominant tree species, presence or absence of dense understorey, basal area of trees, etc)? Did you use 3 flux chambers per site or per plot (i.e. more per site)? How were the chambers arranged in plots (e.g. distance from large trees, understorey vegetation, depressions/mounds, etc)?

We thank the reviewer for requesting more details regarding the flux measurements. We installed one plot in a mixed forest site in the montane forest. There we used seven chambers for the short-term campaigns and three chambers in the first year of long-term measurements. In the second year of the long-term measurements, we increased the number of chambers to five. In the lowland

forests of Yoko and Yangambi, we installed two plots in each site, one in a mixed forest and one in a mono-dominant forest, where more than 60% of the basal area consists of the species *Gilbertiodendron dewevrei.* For the short-term campaigns, we used four chambers in the mixed forest and three chambers in the mono-dominant forest. We started the long-term campaign with four chambers in the mixed forest and two chambers in the mono-dominant forest and after a year, we proceeded with five chambers in the mixed forest and stopped sampling in the mono-dominant forest. The chambers were randomly placed between trees and we avoided hills and depressions. We have now added more information about the plot distribution per site in the paragraph of the study site description (p. 4, L. 11-13) and specified the numbers and placement of soil flux chambers in the subsection of "Soil $CO_2$ flux measurements" (p. 5, L. 7-14). Moreover, we included all information available (basal area, dominant species) in the study site description (p. 3, L. 28-29; p. 4, L. 1-4).

One more note on the number of replicates for CO2 flux measurements: This is not 100% clear from the authors' description, but if I understand correctly, only 3 flux chambers were installed per study site. This is critical because spatial heterogeneity of soil respiration has been described in numerous previous studies, and this could lead to under- or overestimation of soil flux estimates. However, there are a couple of points that the authors could use to address this shortcoming: first, they have measured soil CO2 flux not only in one but in 3 lowland rainforests, and they could look at the difference between sites to describe spatial heterogeneity in the region. Second, if the flux chambers were always installed following a similar scheme, e.g. always at a fixed distance from trees, they would still be comparable even if not 100% representing absolute fluxes. Third, data on GHG fluxes from Africa are very scarce, and one of the reasons is the difficulty in getting research material into or out of the respective countries. I know from personal experience that it can be very difficult to buy or import even simple building material to construct flux chambers, and shipping of environmental samples can be complicated and often requires a lot of paperwork. I can imagine that the situation in DRC might have been similar. Therefore, for future studies on GHG fluxes in regions that are not easily accessible, I recommend the use of the gas-pooling technique by which gas samples from multiple (usually 3-6) chambers are put into the same GC vial, which can help to cover spatial heterogeneity while at the same time reducing the total number of samples. Nevertheless, even if the number of replicates is low and this probably introduces some uncertainty, this information on the magnitude of fluxes and their dynamics is still highly valuable, and I therefore still recommend the study for publication in BG.

We thank the reviewer for acknowledging the difficulties working in remote places and the related compromises sometimes to be made. We realize that our description of replication is lacking important details. As stated above, we used a minimum of three chambers per site. However, for the short-term campaigns, we used up to seven chambers per site. The results from these short-term campaigns showed relatively low variability between chambers at the same site (C.V. of 23% in the lowland forest and 18% in the montane forest) and thus we decided to reduce the number of chambers due to the limited number of evacuated vials for gas sampling (four were used per chamber per sampling for this study). Additionally, the reviewer correctly acknowledged that we measured $CO_2$ fluxes at three different lowland forest sites, which are separated by more than 100 km. The average fluxes of these three sites were also similar (inter-site C.V. of 29%) which gave us further confidence that even those periods where only three chambers were used are representative. We have added these variability statistics to the manuscript (p. 8, L. 6-9). As the reviewer points out already, it was not easy to get materials into the DRC. Therefore, we were only able to increase replication with additional visits to the sites, taking more material to the DRC. Ideally, this would have been done from the start but was not possible due to logistical constraints. Moreover, we appreciate the suggestion to use gas-pooling and will consider adopting this technique in the future.

p. 5 d13C measurements L24-25: wouldn't drawing 3 analytical samples of 20 ml each from the headspace of a 110 ml vial create an underpressure? How did the authors address this?

When withdrawing the samples from the 110 mL vials, a luer-stopcock between syringe and needle was used to avoid underpressure problems when removing the needle from the vial headspace during subsampling. That is, after withdrawal of 25 mL of sample, the luer-stopcock valve between needle and syringe was closed and the syringe was removed from the headspace. After, the plunger was pushed to 20 mL before opening the valve and injecting the subsample to 20 mL Labco vials. This procedure was repeated 3 times. The precision of three analytical replicates was excellent, with a maximum standard deviation of 0.25‰. We have now added a detailed description to the manuscript (p. 6, L. 17-20).

L31: how many litter traps were installed per site, and how were they arranged? L33: how many soil samples were collected per site? What was their arrangement (e.g. distance to chambers, distance to trees, etc)?

Eight litter traps were installed per site and arranged in two rows of four. There was a distance of eight meters between traps. Soil samples were collected at three random positions at each site. We thank the reviewer for their attention to detail and have added this information to the manuscript (p. 6, L. 27-30).

Statistics L6: you assumed little year-to-year variability of your data, but did you actually check if the climatic conditions (rainfall, temperature, moisture) varied between years?

We compiled the flux, temperature and WFPS data in weekly bins for easier presentation of the seasonality of the fluxes. However, for statistical analysis (influence of soil temperature and soil moisture on soil $CO_2$ fluxes) we used the individual fluxes with the actual soil temperature and soil moisture conditions during each flux measurement. As a result, the weekly bins did not affect the results of the statistical analysis. We clarified this issue in the description of the statistical analyses (p. 7, L. 6-8).

p. 6 Figure 2: there are two dips in WFPS in March and October in the lowland forest, where WFPS dropped rapidly from c. 30 to 20%, and then recovered within a week or so – do you have an explanation for this?

The two observed dips are located right before the peak rainy season, therefore, speedy recovery of soil moisture can be expected.

p. 7 Table 1: I know that the R output of lmer looks like this, but it's not very convenient for the reader to understand the results of the statistical analysis. For example, for d13C there is a significant effect of "Montane forest – stream CO2". This is ambiguous: does it mean that the d13C of stream-CO2 is different in montane and lowland forests? Or that the d13C of stream- CO2 is different from the d13C of the other compartments (soil CO2, litterfall, SOC) only in the montane but not the lowland forest? Please use a different way to present these results as they are critical. For example, as a start you could add letters/starts to Figure 2, presenting sign. differences between compartments via different letters, and differences between forests via stars (or something like that).

We agree with the reviewer that we can improve this presentation, to ensure unambiguous interpretation. The p-values (as well as $R^2$s) in the case of linear mixed effect models are only estimations of p-values, and should be interpreted with caution either way, hence the interpretation of the table was mainly meant for the effect sizes. We have remade Figure 3 and included the model information in a new Figure 3 (p. 10). For the ease of interpretation, we split the modeling up in two parallel analyses (one for lowland, one for montane). This now avoids confusion with the

interpretation of too many interaction effects, but the effect sizes of both models will still allow the reader to interpret both inter and intra-forest type effects on $\delta^{13}C$. We also added a new supplementary figure (Figure A4), where the effect sizes of the different $\delta^{13}C$ compartments during the seasons are presented. We thank the reviewer for making this clear, it is in our best interest that the readership of the paper can easily interpret the data we show.

L4-6: You state that stream-CO2 was significantly depleted in the wet season in lowland forests but not montane forests. However, in Table 1 you state "montane forest – wet season – stream CO2" to be significant. Isn't this contradictory (or just another example of how Table 1 could be misinterpreted)?

We thank the reviewer for their attention to detail. In fact, the new parallel analysis of both forest types separately showed that both biomes show a significant depletion of $\delta^{13}C$ – Stream $CO_2$ during the wet season. We amended this in the result section (p. 9, L. 5-8) and added a new Figure A4.

Discussion p. 8 L5-7: Move this to the results section.

Thank you for the suggestion, we have now moved this sentence to the result section (p. 8, L. 6-9).

p. 10 L9-12: Careful, while it is true that with increasing dry season length soil CO2 fluxes might decrease, but it is not clear how future more erratic rainfall patterns and the corresponding more extreme drying-rewetting events will affect respiration, and whether potential CO2 pulses after rewetting compensate or outweigh reduced soil respiration.

We thank the reviewer for providing this qualification and have rewritten this statement to reflect a larger degree of uncertainty, for example, that there could also be $CO_2$ pulses that compensate for lower respiration as a result of these extreme drying-rewetting events (p. 11, L. 18–20).

L25-30: Good call! I agree that the correlations between soil CO2 flux and temperature in tropical systems that show very little annual variation should be handled with care. In your case, they might be significant simply because your sample size is large enough, but I would not over-interpret them. As you correctly state, moisture and C availability are likely the bigger players here.

We are glad that the reviewer agrees with our conservative interpretation.

p.11 L1-4: soil moisture not only controls O2 diffusion but also the diffusion of C substrates to soil microorganisms. Therefore, the response of respiration to moisture is more often an effect of C limitation (at low moisture) than an effect of O2 limitation (which really only becomes critical at very high moisture contents). Please add this to the discussion, and I recommend these papers on the mechanisms underlying this: Manzoni S, Moyano F, Kätterer T, Schimel J (2016) Modeling coupled enzymatic and solute transport controls on ecomposition in drying soils. Soil Biol Biochem 95:275–287. doi: 10.1016/j.soilbio.2016.01.006 Moyano FE, Manzoni S, Chenu C (2013) Responses of soil heterotrophic respiration to moisture availability: An exploration of processes and models. Soil Biol Biochem 59:72–85. doi: 10.1016/j.soilbio.2013.01.002 Moyano FE, Vasilyeva N, Bouckaert L, et al (2012) The moisture response of soil heterotrophic respiration: Interaction with soil properties. Biogeosciences 9:1173–1182. doi: 10.5194/bg-9-1173-2012

We thank the reviewer for this nuanced perspective and have integrated the point along with the mentioned references into the discussion. The sentence now reads: "Soil moisture can influence soil respiration physically and biologically. Physically, soil moisture can limit the transport of C substrate to soil microorganisms (at low soil moisture conditions) and the diffusion of gases through soils,

including both oxygen required for aerobic respiration and respiratory $CO_2$ (in high soil moisture conditions)." (p.12, L. 10-13).

L20: you mention photosynthesis, yet this was not measured and is therefore a bit speculative.

We agree with the reviewer that using the term "photosynthesis" here is a bit misleading in the sense that it indeed does sound this parameter had been measured. We have now re-written this section to avoid the term photosynthesis: "In this study, the link between C assimilation and soil $CO_2$ is evident through […]" (p. 12, L. 30).

p. 12 L4: which canopy processes other than photosynthesis could those be? Furthermore, how do you think that vegetation composition might affect d13C, and could this explain differences between lowland and montane forests? Can different trees have different leaf d13C signatures, which could be reflected throughout the C cascade?

These are two very good points raised by the reviewer. We were mainly thinking of stomatal conductance as the other important canopy process determining $^{13}C$ discrimination. Since there is an interplay between photosynthesis and stomatal conductance on $^{13}C$ discrimination, we lumped these two processes together (as canopy processes). This definition of canopy processes has been added (p. 13, L. 22-23).

The effect of altitude on $\delta^{13}C$ of canopy leaves is well known (Körner et al., 1988, Hultine & Marshall, 2000; Chen et al., 2015) and can be explained by a combination of factors and the two consistent patterns associated with increasing elevation are a decrease in atmospheric pressure and in temperature. The decrease in $O_2$ partial pressure and temperature supposedly promotes a decline in ci/ca and the direct implication of this decline is that $\delta^{13}C$ values become less negative (Wang et al., 2017). We have now added this information to the manuscript (p. 13, L. 9-16).

L6: what are the mechanisms underlying the enrichment of 13C at lower temperatures?

In the subsequent sentence, we explain that temperature changes can result in shifts in microbial communities, which can impact fractionation during heterotrophic soil respiration (Andrews et al., 2000). We have now also added more information about the underlying mechanisms to the manuscript (p. 13, L. 23-24).

Conclusions This is mostly a repetition of the results. Please instead give the "message of the story" – what are the implications of the results you found? What are questions that remain open? And what have we learned?

We agree with the reviewer and rewrote the conclusion section to focus more on the implications of the study results and the remaining research questions.

L24: how were the sites different in vegetation composition? Please describe in the M&M section and also address in discussion

We added information on the vegetation composition to the site description (p. 3, L. 28-29 and p. 4, L. 1-3). Although the vegetation composition may have an effect on rooting density and carbon uptake (net ecosystem uptake) we refrain from elaborating in the discussion as we lack detailed information and would prefer to avoid speculative statements.

L27: what does this indicate, that that there was no temperature dependency of soil respiration between sites?

As our results suggests that respiration in the lowland forests is substrate limited, we reason that the higher temperatures, compared to the montane forest, will not result in an increased soil $CO_2$ flux. We amended this sentence and clarified it in the manuscript (p. 14, L. 15-19).

p. 13 L4: you conclude the paper with the statement that these forests might become C sources under a warming climate, yet you did not find a strong effect of temperature! Instead, you could state that changes of C balance might happen in response to more erratic rainfalls and weather extremes.

We thank the reviewer for pointing us to this important contradiction.
We have rephrased the final statement in our conclusion in a manner that it better fits our observations (p. 14, L. 28-30).

Appendix A: Method supplement L6-25: please use the past tense throughout this section.

We modified the method supplement to the past tense (p. 15, L. 3-21).

p. 16 Figure 3A: change x-axis labels of panel d to the format HH:MM (e.g. 10:00, 15:00,…) to make it clear that those are hours.

We changed the axis label of the d panel (Fig A3, p. 18).

Technical corrections: p. 5 L30: please correct "…during the wet season from October to May"

This has been changed. (p. 8, L. 4)

p. 6 L9: please correct "values were found" (use past tense throughout the results section)

The results section have been modified to the past tense. (p. 8, L. 17)

L8 and elsewhere: You very often use the term "respectively"; however, I'm not a big fan of it, for two reasons: first, sentences become very complex and sometimes hard to understand when using this term, and second, it forces the reader to jump back and forth between the end and the start of the sentence, which disrupts the flow of reading. Very often, you'll find that your sentence won't actually become any longer if instead of using "respectively", you describe the results one after the other, in this case, this could be "The mean [instead of "average", see my earlier comment] annual values we measured in this study in the Congo basin, which are 3.83 _mol m-2 s-1 for the montane forest and 3.69 _mol m-2 s-1 for the lowland forest, are within the range of reported values from other tropical forests." I propose that you revise the MS and try to reduce the use of "respectively". This will make the paper easier to follow.

We very much agree with this comment and have rephrase those sentences throughout the manuscript.

L14: please rearrange "…and they were rather low compared to our flux rates"

The sentence was rearranged. (p. 10, L. 15)

p.9 L2: "…showed marked seasonality [comma] with a 34 % decrease during the dry season [comma] whereas.."

We have added the commas. (p. 11, L. 8)

L4: please rephrase "however, the decrease they found was not as pronounced as…"

This sentence has been rephrased (p. 11, L. 10).

p. 10 L22: "statistically significant correlation"

This has been corrected (p. 11, L. 30).

L18: please rephrase "play a crucial role in controlling soil respiration"

We added "controlling" (p.11, L. 26)

p. 11: L4: please add a comma here, otherwise the phrase is misleading: "stress soil microbial communities, and autotrophic respiration"

A comma has been added. (p. 12, L. 14)

L10: please rephrase as this is otherwise misleading "While soil respiration in lowland forests is most likely C-limited, respiration in montane forests seems to be more sensitive to environmental conditions and could represent a potentially large C source with climate change."

This sentence has been rephrased (p. 12, L. 21-22).

p. 12 L17: enrichment does not occur in the "location" but in the movement from one compartment to another. Please rephrase "the highest enrichment occurs in the last step from soil to stream-dissolved CO2".

This sentence has been rephrased (p.14, L. 4).

L25: please rephrase: "However, in contrast to the lowland forest, the montane forest site exhibited strong seasonality of soil respiration, primarily driven by WFPS during the dry season."

We have rephrased these sentence (p. 14, L. 15-19).

[revised manuscript text omitted]

---

## Referee Report (RR1)

The authors did a good job incorporating the suggested edits. I think the manuscript is now much clearer, the research background and aims are well described. The number of replicates used to measure soil respiration was a bit low, but the authors addressed this by considering spatial variability between study sites, which IMO suffices to make this study more robust. And given the fact that working in the Democratic Republic of the Congo is challenging due to multiple reasons (material import, sample export, safety, site accessibility, etc.), I think they did the best job they could given the circumstances. Furthermore, the authors have revised the discussion to make it less speculative and have pointed out points that need further research, and they have re-written the conclusions, which are now better and pointing forward.

Overall, I am satisfied and recommend this manuscript for publication.